Method

# Combinatorial prediction of marker panels from single-cell transcriptomic data

Conor Delaney[1,†], Alexandra Schnell[2,†], Louis V Cammarata[3,†], Aaron Yao-Smith[4], Aviv Regev[5,6], Vijay K Kuchroo[2,6] & Meromit Singer[1,6,7,*]

## Abstract

Single-cell transcriptomic studies are identifying novel cell populations with exciting functional roles in various *in vivo* contexts, but identification of succinct gene marker panels for such populations remains a challenge. In this work, we introduce COMET, a computational framework for the identification of candidate marker panels consisting of one or more genes for cell populations of interest identified with single-cell RNA-seq data. We show that COMET outperforms other methods for the identification of single-gene panels and enables, for the first time, prediction of multi-gene marker panels ranked by relevance. Staining by flow cytometry assay confirmed the accuracy of COMET's predictions in identifying marker panels for cellular subtypes, at both the single- and multi-gene levels, validating COMET's applicability and accuracy in predicting favorable marker panels from transcriptomic input. COMET is a general non-parametric statistical framework and can be used as-is on various high-throughput datasets in addition to single-cell RNA-sequencing data. COMET is available for use via a web interface (http://www.cometsc.com/) or a stand-alone software package (https://github.com/MSingerLab/COMETSC).

**Keywords** cell types; computational biology; data analysis; marker panel; single-cell RNA-seq

**Subject Categories** Chromatin, Transcription & Genomics; Computational Biology

**Mol Syst Biol. (2019) 15: e9005**

## Introduction

Single-cell transcriptomic technologies have enabled the exciting discovery of novel cell populations within various *in vivo* contexts (Paul *et al*, 2015; Satija *et al*, 2015; Baron *et al*, 2016; Shekhar *et al*, 2016; Singer *et al*, 2016; Villani *et al*, 2017; Jia *et al*, 2018; Kernfeld *et al*, 2018; Vento-Tormo *et al*, 2018; Chapuy *et al*, 2019; Kurtulus *et al*, 2019; Spallanzani *et al*, 2019). Following the discovery of a new cell population of interest based on full transcriptome analysis (of typically a few thousand genes), follow-up studies require succinct gene marker panels by which the cells of interest can be distinguished from the general cell population (Fig 1). For example, isolation of the cell population of choice by flow cytometry assay enables advancement from initial identification via high-throughput transcriptomics to comprehensive functional studies and enables validation of the transcriptomic observations with independent methods.

While the identification of succinct marker panels is a critical step in the transition from initial identification of a cell population to functional exploration and characterization, current techniques used in the literature for the identification of candidate marker panels are substantially limited because they rely on statistical tests designed for other purposes (such as gene differential expression), do not consider gene combinations, and rely on extensive manual curation. A broadly used technique for candidate marker panel annotation from single-cell RNA-seq data consists of generating a ranked list of genes based on their upregulation in the cluster of choice and/or expression fold-change estimates (Paul *et al*, 2015; Satija *et al*, 2015; Baron *et al*, 2016; Shekhar *et al*, 2016; Kernfeld *et al*, 2018; Vento-Tormo *et al*, 2018; Kurtulus *et al*, 2019; Luecken & Theis, 2019). Extensive manual curation is then required to evaluate genes at the top of the list for their ability to provide good classifiers and for their ability to pair with each other to enable favorable multi-gene marker panels (we use the phrase "favorable panel" throughout this manuscript to describe panels that are expected to

1   Department of Data Sciences, Dana-Farber Cancer Institute, Boston, MA, USA
2   Evergrande Center for Immunologic Diseases and Ann Romney Center for Neurologic Diseases, Harvard Medical School and Brigham and Women's Hospital, Boston, MA, USA
3   Department of Statistics, Harvard University, Cambridge, MA, USA
4   Department of Computer Science, Cornell University, Ithaca, NY, USA
5   Department of Biology and Koch Institute of Integrative Cancer Research, Howard Hughes Medical Institute, Massachusetts Institute of Technology, Cambridge, MA, USA
6   Broad Institute of MIT and Harvard, Cambridge, MA, USA
7   Department of Immunology, Harvard Medical School, Boston, MA, USA
    *Corresponding author. Tel: +1 617 632 5134; E-mail: msinger@ds.dfci.harvard.edu
    †These authors contributed equally to this work

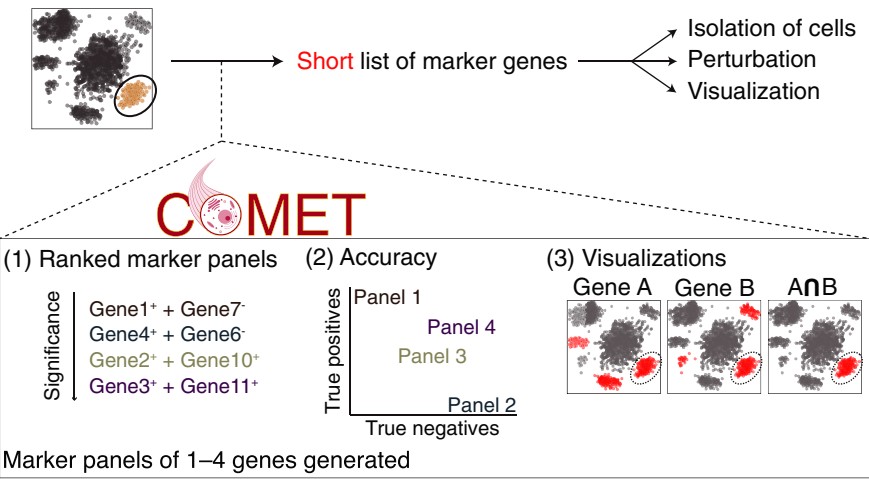

**Figure 1. The COMET framework objective and output.**

Following the identification of a cell population of interest from single-cell high-throughput data (e.g., single-cell RNA-seq), COMET provides a ranking of favorable single- and multi-gene marker panels along with useful statistics and visualizations. The identification of marker panels for a population of interest is important to conduct follow-up functional studies such as isolation, visualization, and perturbation of the population.

achieve good accuracy). A substantial limitation in the use of such techniques is that they do not directly test for a gene's ability to isolate a given cell population from a background, but rather assess the extent to which the gene's expression landscape is significantly different from the given background. We show the limitations of methods testing for upregulation rather than classification potential in Fig 3.

The construction of successful marker panels frequently requires utilizing expression information of multiple proteins or mRNA molecules. The identification of multi-gene marker panels is used for the identification and/or isolation of various cell types (e.g., CD3$^+$CD4$^+$ will identify CD4$^+$ T cells) and cellular subtypes (e.g., CD3$^+$CD4$^+$CD44$^-$CD62L$^+$ will identify naïve CD4$^+$ T cells). Importantly, the genes constructing a successful multi-gene panel may or may not be favorable as single-gene markers (Appendix Fig S1A).

While it is essential to enable the identification of multi-gene marker panels from single-cell RNA-seq data, development of computational tools that provide useful guidance to researchers (e.g., by producing a ranked list of candidate marker panels) is difficult for several reasons. Notable hurdles include the scale and hardness of the algorithmic problem (Appendix Fig S1B and Materials and Methods) and limited availability of experimental reagents for various purposes (e.g., antibodies for flow or *in situ* staining, probes for FISH). The latter requires that a marker panel prediction framework be broad by suggesting multiple (ranked) candidate marker panels to the user, to be assessed for reagent availability and accuracy. Nonetheless, the need within the community to transition from exciting observations at the high-throughput single-cell RNA-seq level to functional, visualization, and perturbation efforts calls for the development of a computational framework which mitigates the challenges and generates an informative ranking of candidate multi-gene marker panels.

In this work, we introduce COMET (COmbinatorial Marker dEtection from single-cell Transcriptomics), a computational framework to identify candidate marker panels that distinguish a set of cells (e.g., a cell cluster) from a given background. COMET

implements a direct classification approach for single genes and utilizes its unique single-gene output to generate exact and/or heuristic-derived predictions for multi-gene marker panels. We show that COMET's predictions are robust and accurate on both simulated and publicly available single-cell RNA-seq data. We experimentally validate COMET's predictions of single- and multi-gene marker panels for the splenic B-cell population as well as splenic B-cell subpopulations by flow cytometry assay, showing that COMET provides accurate and relevant marker panel predictions for identifying cellular subtypes. COMET is available to the community as a web interface (http://www.cometsc.com/) and open-source software package (https://github.com/MSingerLab/COMETSC). We conclude that COMET is an efficient and user-friendly tool for identifying marker panels to assist in bridging the gap between transcriptomic characterization and functional investigation of novel cell populations and subtypes.

# Results

### The COMET algorithm

To identify single- and multi-gene candidate marker panels from high-throughput single-cell RNA-seq data, we developed the COMET framework. COMET takes in as input (i) a gene-by-cell expression matrix (raw counts or normalized), (ii) a cluster assignment for each cell, (iii) 2-dimensional visualization coordinates (e.g., from UMAP, for visualization of plotting), and (iv) an optional input of a gene list over which to conduct the marker panel search, and outputs a separate directory for each cluster that includes ranked lists of candidate marker panels (a separate list for each panel size) along with informative statistics and visualizations (Appendix Fig S2A).

COMET implements the XL-minimal HyperGeometric test (XL-mHG test) (Eden *et al*, 2007; preprint: Wagner, 2015a) to binarize gene expression data in a gene-specific and cluster-specific

manner, assessing for each gene $G$ and cluster $K$, the extent to which gene $G$ could be a good marker for cluster $K$. For each gene, $G$, given a cell cluster of choice, $K$, we use the XL-mHG test to determine an expression threshold by which to binarize the expression values of $G$, such that enrichment of cells from $K$ is maximized (Fig 2A, Appendix Fig S2B, and Materials and Methods). Expression values above the threshold will be set to 1 (the gene is considered "expressed" to a sufficient extent in the cell), while values below the threshold will be set to 0 (the gene is considered "not expressed" in the cell). Genes are also tested for their potential to be used as negative markers in this framework by conducting the above analysis on a gene $G$'s negated expression (Materials and Methods).

We opted to use the XL-mHG test because it is a non-parametric, rank-based test for gene enrichment that has desirable properties for

the purpose of marker discovery. First, the XL-mHG test does not make hard distributional assumptions on the gene expression data (unlike, e.g., likelihood ratio tests applied in hurdle or zero-inflated Poisson models which make linearity assumptions). Second, requirements on the specificity and sensitivity achieved by a marker gene can be tuned within the XL-mHG framework using the X and L parameters to control the minimum number of true positives (X) and the maximum number of false positives (L-X) for a selected threshold (preprint: Wagner, 2015a). We set the default parameters for X and L within COMET to be 15% of the size of the cluster of interest and twice the size of the cluster of interest, respectively (Materials and Methods). The optimal values for these parameters may change in different contexts, and they can be easily set by the user (setting X to 0 and L to the number of cells results in no

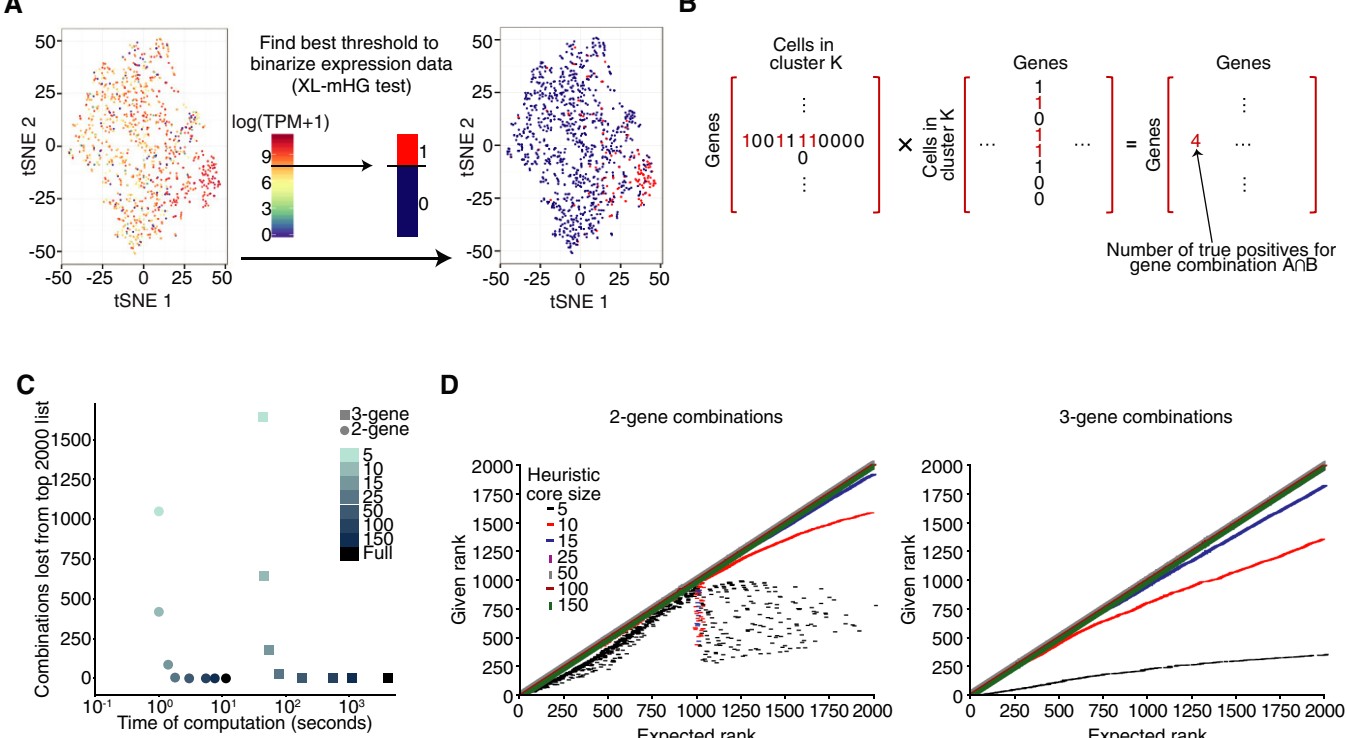

**Figure 2.  Attributes and performance of the COMET Algorithm.**

A    An illustration of the binarization procedure applied by COMET to each gene in a cluster-specific manner via the non-parametric XL-mHG test (preprint: Wagner, 2015a). For each gene, an expression threshold of maximal classification strength for the given cluster is annotated with the XL-mHG test. The XL-mHG P-value measures the significance of the chosen threshold index. This threshold index is then matched to an expression cutoff which is used to binarize gene expression values.

B    The assessment and ranking of multi-gene marker panels by COMET utilize matrix multiplications. Following the binarization of gene expression at the single-gene level, the (i) true-positive and (ii) false-positive rates for all gene combinations considered can be derived from two matrix multiplications (Materials and Methods). Illustrated here is the matrix multiplication to annotate true-positive rates for 2-gene marker panels. The true-positive and false-positive values are then used to compute hypergeometric enrichment P-values for all pairs.

C, D    Marker panel predictions by COMET align closely between the heuristic approach and exact computation. Results displayed are a representative example, computed from analysis COMET's performance when analyzing the follicular B-cell cluster (Fig 5A). (C) Running time can be improved with a proper choice in heuristic core size to maintain accuracy of results. The number of missed combinations in the top 2,000 ranked combinations is plotted against the time of computation for the 2-gene and 3-gene cases for a variety of heuristic core sizes to determine accuracy versus runtime. The leveling off of the number of missed combinations provides a good place to set the heuristic core size for best speed-up and accuracy; COMET's current default is 50. (D) The COMET generated rankings of each of the top 2,000 combinations for 2-gene (left) and 3-gene (right) panels are plotted against each combination's ranking from COMET's heuristic approach when using different sizes for the gene set heuristic core (Materials and Methods). At a core size of 25 (using the top-ranking 25 single genes as the heuristic core) or larger, results align very closely between the heuristic and exact approaches.

restrictions on the true-positive and false-positive rates, and is the original mHG test). COMET outputs a ranked list of candidate single-gene markers by integrating the XL-mHG *P*-values and the log fold change of gene expression into a simple scoring metric (Materials and Methods). In addition to this ranking, COMET provides the true-positive and true-negative rates for each marker candidate and generates an informative plot by which to identify genes that are outliers in their specificity-to-sensitivity ratio (Appendix Fig S2A).

Construction of marker panels that include multiple genes is frequently required to isolate/identify a cell population of interest with high precision. We opted to utilize the binarization in expression achieved for each gene (in a cluster-specific manner) by the XL-mHG test to assess the performance of multi-gene marker panels in isolating the cell cluster of interest. COMET computes for all gene combinations of a given size (of 2–4 genes) the true-positive and true-negative rates the given combination would achieve by using matrix multiplications on the binarized expression matrices (Fig 2B and Materials and Methods). This matrix-multiplication procedure enables an efficient computation of the needed statistics across all possible gene combinations for a given marker panel size (computed for combinations of size 2–4 genes), to achieve a ranking of relevant multi-gene candidate marker panels. A ranking of the candidate multi-gene marker panels is done based on enrichment of cells expressing the entire gene panel in the cell cluster of choice (hypergeometric enrichment *P*-value) combined with a "Cluster Clear Score" (CCS) which we define as follows:

$$CCS = \sum_{C \in \mathcal{C} \setminus \{K\}} TN_C^{after} - TN_C^{before}$$

where $TN_C^{before}$ is the true-negative percent in cluster $C$ for the single gene in the panel with the lowest *P*-value when considered as a single-gene marker (the "lead" gene) and $TN_C^{after}$ is the true-negative percent in cluster $C$ for the panel (after addition of the remaining genes in the panel). The CCS measure is an estimate of the extent to which using multiple markers has improved precision as compared to use of any single marker within the panel, and is meant to assist the user in identifying marker panels that significantly improve accuracy when used in combination. COMET outputs a ranked list of candidate marker panels for each marker panel size, along with informative statistics and plotted visualizations (e.g., Appendix Fig S3 for a three-gene panel).

While an exhaustive search is required to ensure obtaining the optimal solution(s) and hence an accurate ranking of candidate multi-gene marker panels (Materials and Methods), such may not be feasible for inputs consisting of many genes (e.g., the entire gene list) and/or many cells. To increase efficiency in computation time such that input size is not a limiting factor, we implemented a heuristic within COMET to rank multi-gene candidate marker panels. The user can opt to use either the exhaustive or heuristic approach based on the number of genes searched across for marker panels (input (4)), the number of cells in the expression matrix (input (1)), and the computational resources available. COMET's heuristic defines a "core" set of genes as the top $N$ genes in the single-gene marker panel ranking and assesses combinations of those core genes with all genes given as input ($N$ is set to 50 by default and can be changed by the user). For a gene marker panel of

size $m$, the heuristic considers all combinations in which at least $(m-1)$ genes are from the core set (Materials and Methods; for the 4-gene case, all of the genes in the marker panel will be from the core set). We validated that the rankings generated by our heuristic approach align well with those generated by an exhaustive ranking, while significantly reducing running time (Fig 2C and D).

COMET was designed as a general framework and can be applied to any type of high-throughput data (including non-single-cell data), using any kind of normalization and clustering based on user preference (COMET can also be applied successfully to raw counts). Normalization of the input expression matrix and clustering is left to the user to allow for maximum flexibility. The gene list over which marker panel candidates are assessed can be the entire gene list or a subset of genes (e.g., all surface genes or a list of genes for which favorable antibodies or probes are available). Contrary to other methods that pool information across all genes to infer parameters (e.g., hurdle models or zero-inflated Poisson hierarchical models), COMET treats each gene independently and can therefore be applied to user-specified gene lists as described above.

### COMET is robust and accurate in identifying favorable markers

To test COMET's performance in identifying favorable marker panels, we assessed COMET's performance on Monte Carlo simulations as well as publicly available single-cell RNA-seq datasets. Here, we define a marker to be favorable if it can likely be used to efficiently sort out cells from the cluster of interest via, for instance, flow cytometry assay. We generated Monte Carlo simulations using both synthetic gene expression data (generated using a Gaussian distribution) and synthetic gene counts data (generated using a negative binomial distribution) for one gene in many cells (Fig 3A and B, and Appendix Fig S4). COMET was compared to several gene differential expression (DE) tests frequently used to identify single-gene marker panels (Finak *et al*, 2015; Satija *et al*, 2015; Ntranos *et al*, 2019). Common gene DE tests included in the comparison are Welch's *t*-test, the Wilcoxon rank-sum test (and its generalization the Kruskal–Wallis test), the Kolmogorov–Smirnov test, and the likelihood ratio test on a logistic regression model where cell cluster (1 if the cell belongs to the cluster of interest, 0 otherwise) is regressed against an intercept only or both an intercept and the expression value of the gene in that cell (Materials and Methods). Simulations showed that the COMET procedure detects good markers and discards poor markers regardless of sample size, contrary to other tests whose power increases rapidly with sample size (Fig 3B and Appendix Fig S4). The X and L parameters of the XL-mHG test play an important role in this favorable behavior.

To some extent, the binarization of gene expression implemented in COMET can be related to a classification task. To assess COMET's performance compared to standard classification procedures, we performed Monte Carlo simulations on cell-by-gene expression matrices. Two distinct simulation procedures were used, including a simple Gaussian generative model for gene expression data and a noisy Poisson–Gamma generative model for gene count data (Materials and Methods). Synthetic expression data were generated for two cell clusters (one of which is the cluster of interest) and many genes pertaining to three categories: good markers (genes which separate well the two clusters), poor markers (e.g., markers of cell subclusters, measurement outliers), and

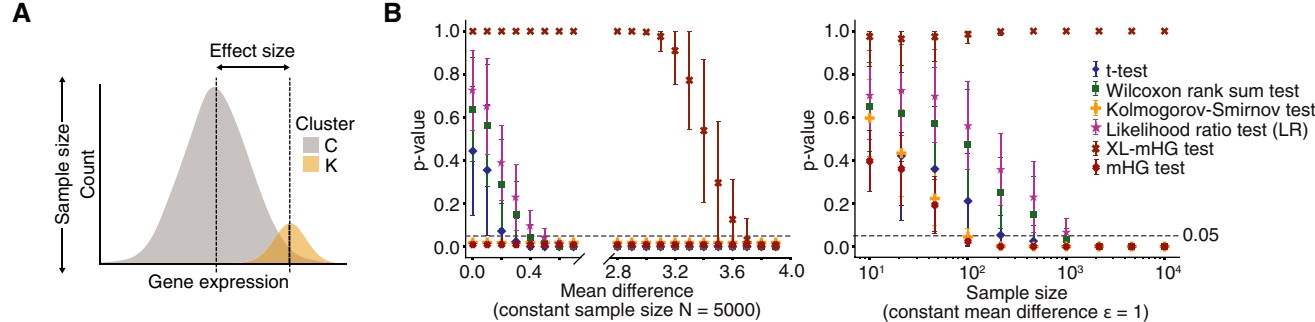

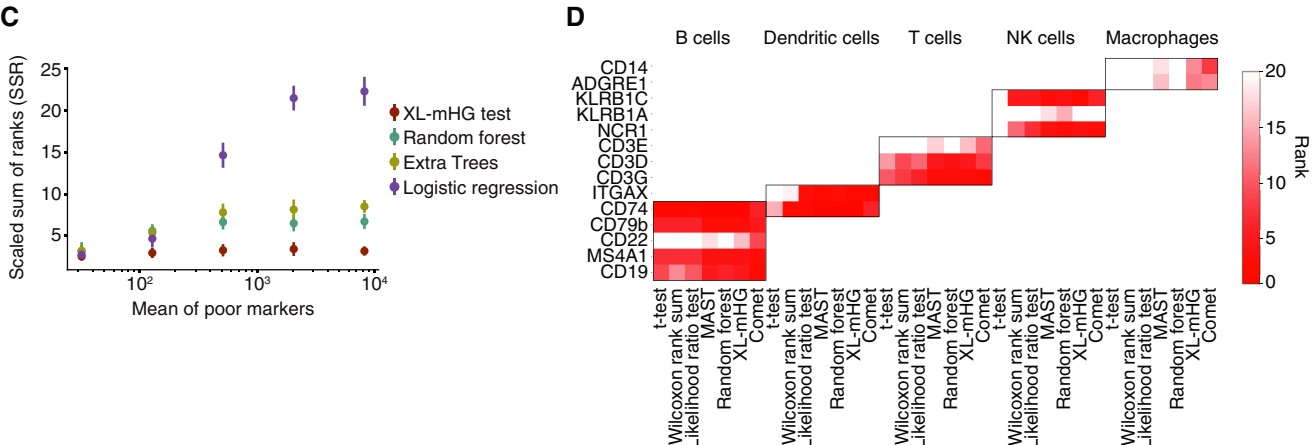

**Figure 3.  COMET accurately and efficiently computes marker panels for cell populations.**

A, B   The XL-mHG test outperforms various differential expression tests in identifying favorable marker genes to be used as markers from simulated datasets (A, Materials and Methods), with respect to both robustness to small effect sizes (B, left) and sensitivity to sample size (B, right). B, left: When varying the magnitude of the difference between the means of the expression distributions for the cluster of interest (K) compared to the background (C) (termed here "effect size", see illustration in (A)), common DE tests drop below 0.05 significance level at small effect sizes (of approximately 0.4), while the XL-mHG test reaches significance only at approximately 3.6. Identification of favorable marker genes requires achieving satisfactory sensitivity and specificity rates which would not be achievable in cases of small effect sizes (due to the large overlap across the compared distributions). Hence, the XL-mHG test performs better than commonly used DE tests in that it does not assign significant *P*-values to genes that are differentially expressed but would be poor markers due to small effect sizes. B, right: When varying the total number of cells simulated in clusters K and C (termed here "sample size", see illustration in (A)) for a fixed and small effect size of 1, common DE tests pick up the small difference in expression as significant once the sample sizes become large (and the detection power increases), while the XL-mHG test does not reach significance and would not consider such genes as potential markers. The small effect size in this example simulates a poor marker for which desirable sensitivity and specificity rates could not be achieved, and this is controlled in the XL-mHG test by the X and L parameters.

C       The XL-mHG test outperforms logistic regression and tree ensemble classifiers (including random forest and extra trees) in identifying favorable genes to be used as markers from simulated datasets (noisy Poisson–Gamma generative model, see Materials and Methods). The scaled sum of ranks (SSR) metric indicates the ability of a method to rank highly good marker genes, with a value of SSR = 1 indicating optimal ranking.

D       COMET accurately identifies established markers for cell subpopulations in mouse spleen. Shown are the rankings of established marker genes for immune populations generated by different methods used for single-gene marker identification. Data are taken from the spleen tissue of the MCA (Han *et al*, 2018).

Data information: Error bars indicate one standard deviation across 100 simulation runs (thresholded below at 0 and above at 1).

non-markers (genes with similar expression across both clusters) (Appendix Fig S5A, Materials and Methods). Recent literature mentions the use of Logistic Regression (LR) in gene DE analysis (Ntranos *et al*, 2019), while Random Forests (RF) have been used for a variety of tasks in genomics (Irrthum *et al*, 2010). We used each of XL-mHG test, LR, and RF to construct a ranking of potential markers and compared the methods' rankings to the optimal ranking (known from the simulation) using the Scaled Sum of Ranks (SSR) metric. We defined SSR to determine the extent to which the good markers are ranked at the top of the list (Materials

and Methods). An SSR score of 1 reflects a ranking in which all good markers are ranked at the top of the list, in higher places than any of the poor markers and the genes with similar expression across clusters. We compared the SSR scores across the LR, RF, and XL-mHG classification methods and observed that poor markers had a detrimental effect on the identification of good single-gene markers by LR and RF, while the XL-mHG test was robust to the quantity and expression rates of poor markers in the data (Fig 3C and Appendix Fig S5). The X and L parameters play an important role in protecting COMET against the selection of

genes which constitute poor markers for the cluster of interest yet enjoy a strong predictive power (such as subcluster markers).

To evaluate COMET's ability to identify novel surface single-gene markers from real data, we evaluated COMET's prediction of cell surface markers for splenic cell populations and focused on the B-cell population due to its abundance and well-established marker set. Single-cell data for the spleen tissue from the Mouse Cell Atlas (Han *et al*, 2018) were processed using COMET for a curated list of murine cell surface proteins (Chihara *et al*, 2018) (a default gene list used by COMET unless specified by the user). We compared the rankings of known single-gene markers obtained by COMET for the spleen data from the Mouse Cell Atlas to other differential expression tests (Welch's *t*-test, Wilcoxon rank-sum test, likelihood ratio test, and MAST hurdled *t*-test). COMET performs well in identifying known single-gene markers for the different cell populations identified in spleen (Fig 3D), serving as a validation that COMET distinguishes well across different cell types for which established markers are annotated. The well-established B-cell markers (Nadler *et al*, 1981, 1983) CD19 and CD20 (Ms4a1) ranked 2$^{nd}$ and 3$^{rd}$ in the COMET output, respectively (Fig 4A–C and Table EV1). Flow cytometry-based assay confirmed that the additional top-ranking candidates Ly-6D and CD79b co-stain well with CD19, confirming the accuracy of COMET's predictions for single-gene marker panels (candidates for validation chosen by antibody availability) (Fig 4C and D). These candidates also showed limited co-staining with known T-cell marker CD3 (Meuer *et al*, 1983), showing their specificity as B-cell markers (Appendix Fig S7). Indeed, COMET predicted the combination of Ly-6D$^+$CD3$^-$ to be a favorable 2-gene marker panel which is validated by flow cytometry staining (Appendix Fig S8). When comparing COMET's performance to that of other methods for identifying single-gene markers, we found that COMET's rankings were slightly higher from that of other methods for the two well-established B-cell markers Cd19 and Ms4a1 (CD20), and slightly higher or comparable with respect to the two markers we validated by flow cytometry (Ly6d and Cd79b) (Table EV1). Having all methods tested be comparable with respect to the identification of single-gene markers for B-cell markers by all methods is expected given that identifying markers for a distinct cell type is a relatively simple task. Our results show that COMET can utilize single-cell transcriptomic data to predict highly specific cell surface markers for cell populations.

### COMET identifies novel marker panels for cellular subtypes

We envision a primary use for COMET in the identification of candidate marker panels for subpopulations of a given cell type. To assess COMET's ability to predict successful marker panels for cell subpopulations, we experimentally validated COMET's predictions for one of four splenic B-cell clusters identified in mouse spleen in the Tabula Muris dataset (Tabula Muris Consortium, 2018) (Fig 5A). Splenic B-cell populations typically include ~65–90% follicular B cells, which are identified by expression of CD23 as a cell surface marker. Additionally, the splenic B-cell population includes 5–15% marginal zone B cells which can be identified by expression of CD21 as a cell surface marker (Pillai *et al*, 2005; Allman & Pillai, 2008). We assigned these two main subtypes of B-cell splenic populations to Tabula Muris clusters 0 (follicular B cells) and 2 (marginal zone B cells) by using the established markers and

subtype-specific gene signatures (Fig 5B and Appendix Fig S9). Focusing on the follicular B-cell cluster, we validated using flow cytometry assay that the most highly ranked single-gene markers predicted in the COMET output (CD55 (rank 1), CD62L (Sell, rank 2), and CXCR4 (rank 3), see Table EV2) co-stain well with the established marker CD23 (ranked 4) (Fig 5C and D, and Appendix Fig S9A and B). When comparing COMET's performance to that of other methods for identifying single-gene markers for the follicular B-cell cluster, we found that COMET's ranking of Cxcr4 was significantly higher than the ranking of any other methods (rankings by other methods ranged from 12 to 53, for Wilcoxon rank-sum test and LRT, respectively, see Table EV2) and was comparable for the other validated markers (Cd55 and Sell). Our results show that COMET can predict with accuracy highly specific surface markers for subpopulations of an established cell type from single-cell transcriptomic data.

An important aspect of identifying marker panels for cell subtypes involves accurate prediction of marker panels that consist of more than one gene, to enable high accuracy of subtype isolation/targeting when a single-gene marker may not be sufficient to predict a cell cluster. We therefore tested COMET's ability to predict accurate marker combinations for the follicular B-cell subpopulation (the specific high-ranking combinations to validate were determined by antibody availability). COMET predicted the combination (CD62L$^+$CD44$^-$) for the isolation of follicular B cells, which ranked 22 with favorable true-positive and false-positive predictions (Fig 6A). We observed that staining of CD62L$^+$CD44$^-$ yields a significantly cleaner population of follicular B cells (defined as CD23$^+$) than CD62L alone (Fig 6C). We tested by flow cytometry a second highly ranked two-gene combination for follicular B-cell identification and observed an improvement in using the combination predicted, CD55$^+$CD62L$^+$, rather than either CD62L$^+$ or CD55$^+$ alone (Fig 6B and D, and Appendix Fig S10). Importantly, the combinations for validation were selected by their COMET ranking as well as by antibody availability. The combinations assessed ranked 22 (CD62L$^+$CD44$^-$) and 38 (CD62L$^+$CD55$^+$). Hence, gene combinations that rank at such levels and have available antibodies are candidates for being good gene marker panels for subpopulations identified by single-cell transcriptomics. Taken together, we have shown that COMET is an applicable tool for the prediction of single- and multi-gene surface marker panels for cell populations and subtypes.

## Discussion

The fast-increasing number of single-cell RNA-seq datasets being generated and analyzed is revealing novel cell types and subtypes in a variety of systems. Propelled by the exciting findings to date, multiple consortia have formed to identify and characterize novel cellular types and subtypes in comprehensive structured efforts (Regev *et al*, 2017; Adlung & Amit, 2018; HTAN: Mapping Tumors across Space and Time, 2019). Advancing from a high-throughput characterization to deep functional studies of such novel populations requires the annotation of succinct marker panels to enable isolation, visualization, and perturbation. In this work, we introduced COMET, a powerful framework to identify succinct marker panels that distinguish a selected subset of cells from a given

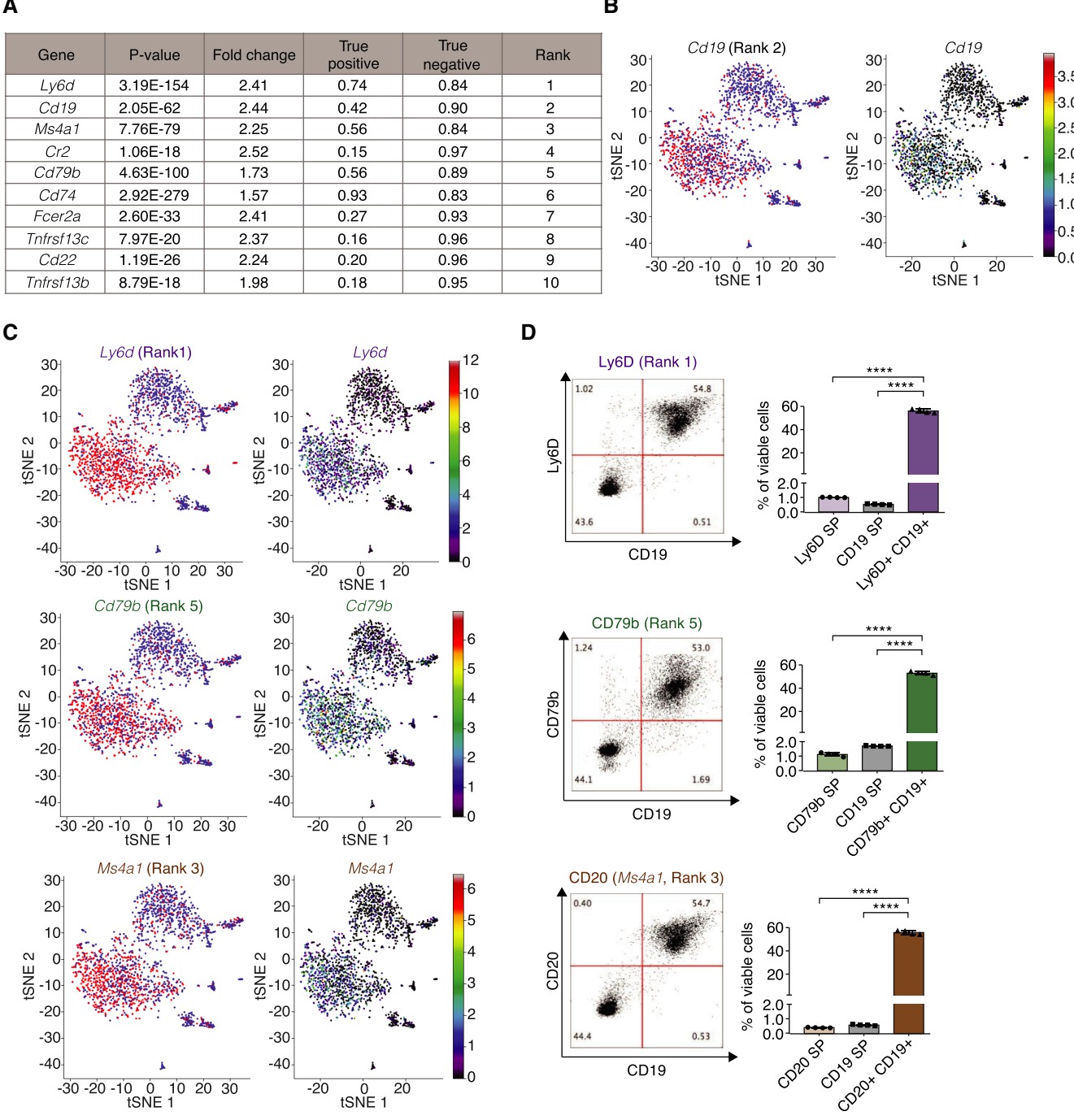

**Figure 4. COMET identifies favorable markers for splenic B cells.**

A–C COMET outputs for the splenic B-cell population from the MCA dataset (Han *et al*, 2018). (A) COMET output of the top 10 ranked candidate marker genes. (B, C) COMET plots the expression of a gene across all cells (right) and the binarized values of gene expression following binarization (red: expressed; blue: not expressed) by the XL-mHG threshold (left). Shown are COMET visualization outputs for CD19 (B) and Ly-6D, CD20, and CD79b (C).

D Flow cytometry analysis comparing the protein level staining of CD19, an established marker for B cells (Nadler *et al*, 1983), with three top-ranking marker genes in the COMET output confirms that COMET's top-ranking candidate markers are favorable for flow cytometry staining of B cells. The genes to validate were selected based on availability of trustable antibodies (SP = single positive). Bars and error bars indicate the mean and standard deviation. ****$P < 0.0001$; $n = 4$ biological replicates; unpaired, two-tailed *t*-test.

background. In difference from currently used techniques for identification of single-gene markers, COMET takes a direct classification approach and generates ranked lists of multi-gene marker panels.

We have demonstrated in this work that COMET can utilize single-cell RNA-seq data to identify favorable marker panels for isolation of cell types and cell subtypes by flow cytometry assay. Importantly,

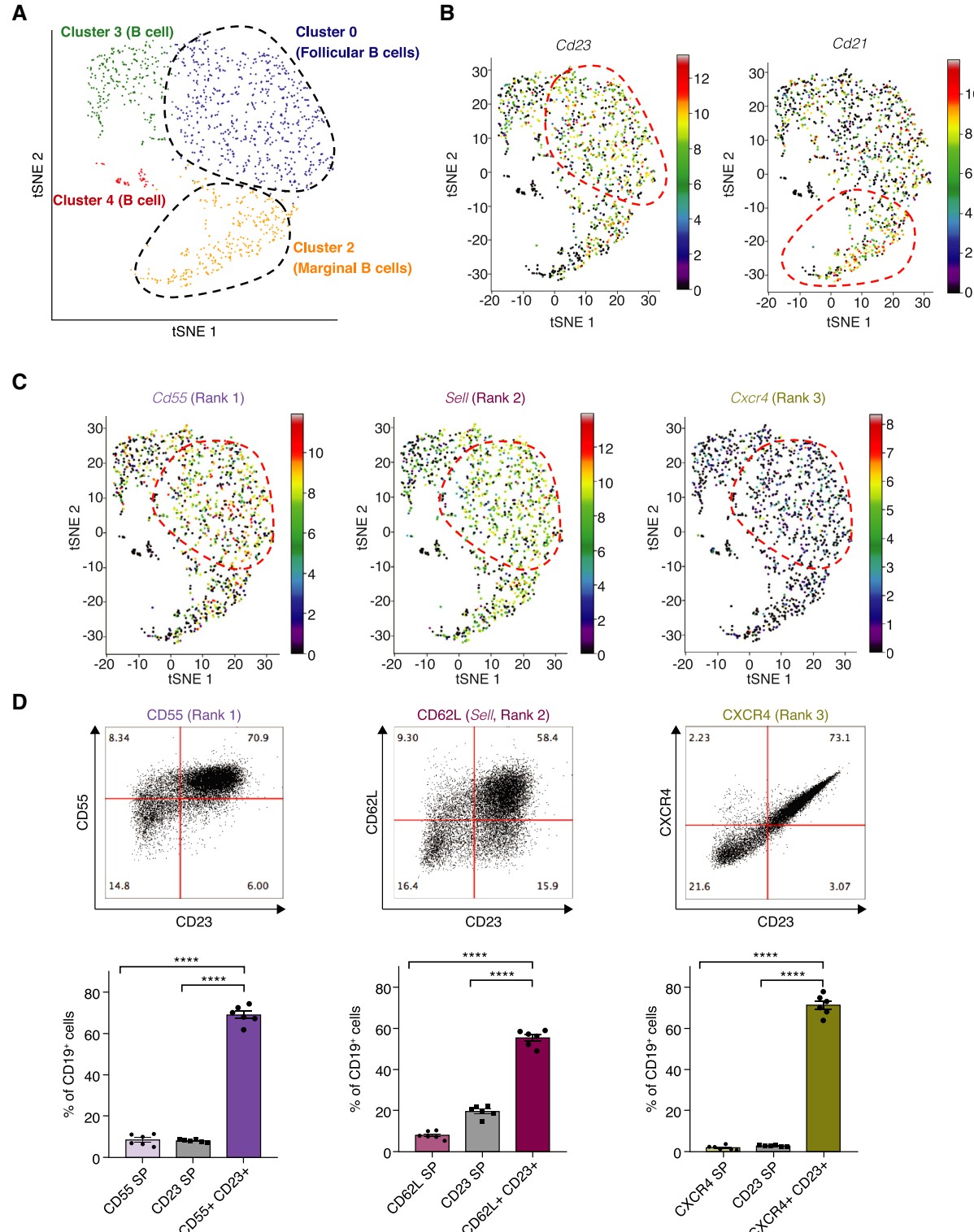

**Figure 5. COMET identifies favorable markers for splenic follicular B cells.**

A Clustering and t-SNE visualization of splenic B cells as generated by Tabula Muris (Tabula Muris Consortium, 2018).

B Expression of follicular B-cell marker CD23 and marginal zone B-cell marker CD21 in the splenic B-cell dataset from Tabula Muris as visualized by COMET.

C Expression of the three top-ranking markers for follicular B-cell output by COMET, CD55, CD62L, and CXCR4, as visualized by COMET.

D Flow cytometry analysis comparing the protein level staining of CD23, an established marker for follicular B cells, with the three top-ranking marker genes in the COMET output confirms that COMET's top-ranking candidate markers are favorable for flow cytometry staining of the follicular B-cell subtype. The genes to validate were selected based on availability of trustable antibodies (SP = single positive). Bars and error bars indicate the mean and standard deviation. ****$P < 0.0001$; $n = 6$ biological replicates; unpaired, two-tailed $t$-test.

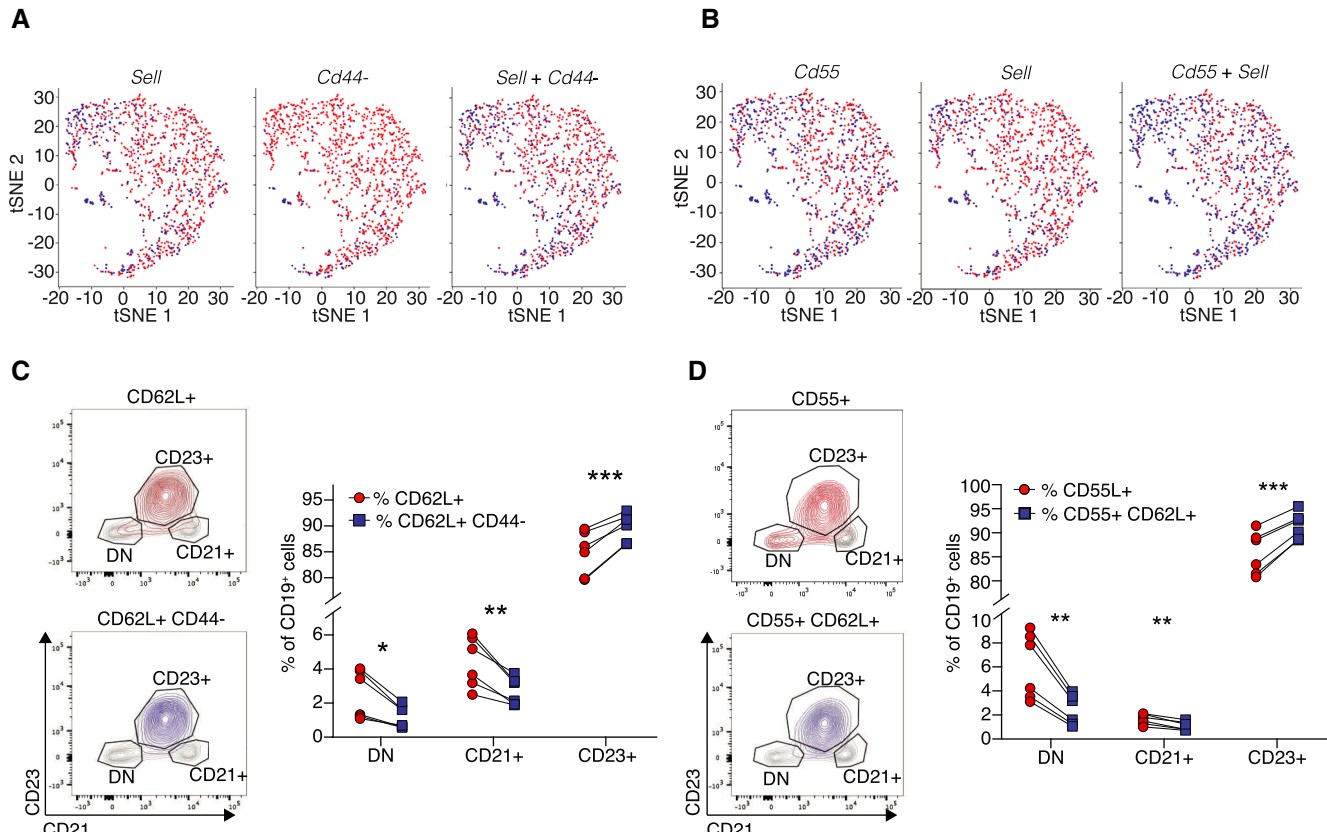

**Figure 6. COMET identifies favorable multi-gene marker panels for splenic follicular B cells.**

A, B  COMET outputs for two highly ranked 2-gene marker panels predicted by COMET to isolate the splenic follicular B-cell population, based on analysis of the Tabula Muris dataset (Tabula Muris Consortium, 2018). Shown are binarized values of gene expression following binarization by the XL-mHG threshold for each gene separately (left, middle) and when using both genes combined (right).

C, D  Flow cytometry staining for the marker combinations CD62L$^+$CD44$^-$ (C) and CD55$^+$CD62L$^+$ (D) confirms that COMET's candidate multi-gene marker panels are favorable for flow cytometry staining of splenic follicular B cells. Both marker combinations included a significantly higher frequency of follicular B cells (CD23$^+$) and a lower frequency of other B-cell subpopulations (DN and CD21$^+$) than the single staining for CD62L$^+$ and CD55$^+$, respectively. (DN = double negative). The marker combinations were selected based on availability of established antibodies. Bars and error bars indicate the mean and standard deviation. *$P < 0.05$; **$P < 0.01$; ***$P < 0.001$; $n = 6$ biological replicates; unpaired, two-tailed $t$-test.

our validation that COMET's predicted marker panels for B cells and B-cell subtypes can be used in flow cytometry assay to isolate the population of choice emphasizes that transcriptomic data can be utilized via COMET to identify favorable marker panels at the protein level.

A main contribution within the COMET tool is the introduction of a framework for identifying multi-gene combinations that constitute favorable marker panels. Although the problem of finding an optimal multi-gene marker panel is computationally intractable (as we prove in Materials and Methods), we introduce within the COMET framework an efficient method to conduct an exhaustive search (for small marker panels, utilizing the efficiency of matrix multiplications) as well as heuristics which scale to larger sizes of marker panels and gene inputs, and are shown to achieve accurate performance. Importantly, the COMET framework is non-parametric for both the single- and multi-gene cases, enabling identification of non-linear relationships when searching for marker panels. COMET is also highly flexible in the gene space within which the search for markers is conducted (e.g., restricting the search space to genes for

which favorable antibodies exist). Additional notable features of the COMET framework are the direct assessment of a gene's classification potential using informative statistics, and the direct assessment of negative markers (genes that are not expressed in the cell cluster of choice) as single markers or as part of a multi-gene panel (validated examples of a panels with a positive and negative marker are seen in Fig 6C and Appendix Fig S8). We observed COMET to work well across a range of technologies and sequencing depths (from an average count of 547 genes per cell in the microwell-seq MCA dataset to an average count of 1,825 genes per cell in the Smart-Seq2 Tabula Muris dataset).

While the COMET framework enables the identification of marker panels from high-throughput transcriptomic data, there are several outstanding challenges that COMET does not currently address. In predicting marker panels from transcriptomic data, an underlying assumption made is that genes correlate well between their transcriptional and protein/cell surface abundance. Due to factors such as mRNA stability, translation efficiency, protein stability, and protein transport, we know that this assumption is not

always accurate. Additionally, antibody quality and specificity can contribute to discrepancies between cellular mRNA levels and surface protein detection rates with flow cytometry. As observed in Fig 5D, the COMET ranks do not always perfectly correlate with performance in flow cytometry (CXCR4 ranked 3[rd] but performs better than ranks 1 and 2). It will be beneficial to incorporate within COMET information regarding the extent to which the transcriptional state of each gene correlates with its protein abundance, as well as availability of validated antibodies. Incorporation of data from, e.g., the Human Protein Atlas (Uhlén *et al*, 2015; The Human Protein Atlas) could assist in this regard. When assessing multi-gene marker combinations, COMET considers the "AND" relationship between genes (cells expressing both gene A and gene B). Addressing "OR" logic between genes (cells expressing either gene A or gene B) and constructing gene panels that incorporate a mixture of "AND" and "OR" relationships across genes will be the subject of future expansions of COMET. Last, exciting technological advances are enabling the incorporation of tens and potentially hundreds of genes to be used within marker panels (e.g., multi-colored flow cytometry, CyTOF, and CITE-seq). Expanding the COMET framework to explore both small and large marker panels would be favorable in this respect.

Along with its broad applicability to single-cell transcriptomics data, the COMET framework can be utilized for other instances by merely changing the input to the available software. The statistical framework implemented in COMET conducts a ranked non-parametric search to identify single- and multi-feature marker panels that best distinguish a specified subset of samples. Hence, the COMET framework and tool can be used to identify marker panels from additional high-throughput datasets such as population RNA-seq data and CpG site methylation data. Importantly, analysis of data from technologies that measure both RNA and protein levels (e.g., Stoeckius *et al*, 2017) can be trivially processed by COMET by providing an input in which each of the measurements (RNA or protein) is a "gene" in the expression matrix. Since each "gene" is assessed within COMET in a non-parametric manner to produce a binarization of the data, protein and RNA measurements can be combined, as well as any other "Omic" datasets. An additional use of the COMET framework can be in the identification of markers for transcriptional programs of interest (rather than cell clusters). Recent works have highlighted the annotation of cellular transcriptional programs (by, e.g., NNMF and topic modeling) as a means of identifying important features of variance within single-cell RNA-seq datasets (preprint: Bielecki *et al*, 2018; Filbin *et al*, 2018; Jerby-Arnon *et al*, 2018). COMET's computational framework could be applied to the identification of marker panels for transcriptional programs or motifs of interest, by assigning cells as expressing or not expressing a given program, and we expect it will be interesting to explore marker annotation in that space.

We anticipate that the use of COMET will propel the transition from novel characterization-focused observations (made via methods such as single-cell RNA-seq) to targeted studies that focus on functional aspects of the identified findings. COMET is available to the community as both a web interface (http://www.cometsc.com/) and an open-source stand-alone Python package (https://github.com/MSingerLab/COMETSC).

# Materials and Methods

## Reagents and Tools table

| Reagent/Resource | Reference or source | Identifier or catalog number |
| --- | --- | --- |
| **Experimental models** | | |
| C57BL/6J, Mus musculus | Jackson Laboratory | Stock number: 000664 |
| **Antibodies** | | |
| Armenian Hamster anti-CD3ε (clone 145-2C11), monoclonal, 1:500 | BioLegend | Cat # 100335 |
| Armenian Hamster anti-CD11c (clone N418), monoclonal, 1:500 | BioLegend | Cat # 117317 |
| Rat anti-CD19 (clone 6D5), monoclonal, 1:500 | BioLegend | Cat # 115529 |
| Rat anti-CD20 (clone SA275A11), monoclonal, 1:300 | BioLegend | Cat # 150403 |
| Rat anti-CD21 (clone 7E9), monoclonal, 1:300 | BioLegend | Cat # 123421 |
| Rat anti-CD23 (clone B3B4), monoclonal, 1:100 | BioLegend | Cat # 101613 |
| Armenian Hamster anti-CD55 (clone RIKO-3), monoclonal, 1:300 | BioLegend | Cat # 131803 |
| Rat anti-CD62L (clone MEL-14), monoclonal, 1:300 | BioLegend | Cat # 104419 |
| Armenian Hmaster anti-CD79b (clone HM79-12), monoclonal, 1:300 | BioLegend | Cat # 132805 |
| Rat anti-CXCR4 (clone L276F12), monoclonal, 1:300 | BioLegend | Cat # 146511 |
| Rat anti-Gr-1 (clone RB6-8C5), monoclonal, 1:500 | BioLegend | Cat # 108407 |
| Rat anti-Ly-6D (clone 49-H4), monoclonal, 1:300 | BioLegend | Cat # 138605 |
| **Chemicals, enzymes and other reagents** | | |
| eBioscience eFluor 506 fixable viability dye | Thermo Fisher Scientific | Cat # 65-0866-14 |

**Reagents and Tools table** (continued)

| Reagent/Resource | Reference or source | Identifier or catalog number |
|---|---|---|
| Gibco ACK lysis buffer | Fisher Scientific | Cat # A1049201 |
| FCS (newborn calf serum) | Sigma-Aldrich | Cat # N4637 |
| DPBS | Gibco | Cat # 14190-250 |
| **Software** | | |
| GraphPad Prism Version 8.1.0 | https://www.graphpad.com/ | |
| FlowJo Version Version 10.5.3 | https://www.flowjo.com/ | |
| Python Version 3.6 | https://www.python.org/ | |
| R Version 3.5 | https://www.r-project.org/ | |
| **Other** | | |
| LSR II | BD Biosciences | |
| LSRFortessa | BD Biosciences | |

## Multi-gene marker identification problem

We prove below that the general problem of identifying an optimal multi-gene marker panel for a cell cluster of interest $K$ is NP-hard. Consider an instance in which any gene combination (e.g., $\{G_i, G_j\}(i, j \in \{1, \ldots, d\})$) satisfies $TP_{combination} = Q$, where TP is the true-positive rate, for some constant $Q \in [0, 1]$. Hence, all gene combinations are equivalent with respect to their true-positive rate, and we need to find the gene combination which maximizes the true-negative rate, restricted by some $k$ (the maximum number of genes allowed in the panel). Let $\mathcal{C} = \{C_1, \ldots, C_d\}$, where $C_i$ is the set of cells outside of the cluster of interest $K$ that do not express gene $G_i (i \in \{1, \ldots, d\})$. We want to find a set of indices of size up to $k$ such that the union over their sets in $C$ has maximal cardinality (in the best case, the cardinality of $K^c$, the complement of $K$), in order to maximize the true-negative rate of the gene combination chosen. Hence, the multi-gene marker identification problem is polynomially reducible to the maximum coverage problem which is NP-hard. Appendix Fig S1 demonstrates the scaling of the number of combinations in increasing marker panel sizes.

## XL-mHG test

We provide a brief description of the XL-mHG test and refer the reader to the relevant references (Eden *et al*, 2007; preprint: Wagner, 2015a) for more extensive details. The minimal HyperGeometric (mHG) test is a rank-based non-parametric test for determining gene enrichment. Consider a gene $G$ expressed in a set of $n$ cells comprising two clusters $K$ and $C$, where $K$ denotes the cluster of interest (in the case of more than one cluster, all clusters but $K$ are merged to form $C := K^c$). Let $\{X_i\}_{1 \le i \le n}$ denote the random variables indicating the expression of $G$ in the cells $i = 1, \ldots, n$, and $\{C_i\}_{1 \le i \le n} \in \{0, 1\}^n$ (where 1 refers to cluster $K$ and 0 to cluster $C$) denote the cluster assignment of each cell. The test starts by sorting the cluster assignments in decreasing order of gene expression, i.e.,

$$S = \left(C_{\pi(1)}, C_{\pi(2)}, \ldots, C_{\pi(n)}\right)^T$$

where $X_{\pi(1)} \ge X_{\pi(2)} \ge \ldots \ge X_{\pi(n)}$. The test is based on the observation that, given a threshold on the sorted vector $S$ that defines the top of this vector, enrichment can be quantified using a HyperGeometric (HG) test for that threshold. Instead of working with a fixed threshold, the mHG test explores all possible thresholds $t \in [0, n]$ and calculates a HG $P$-value $p_{(t)}^{HG}$ for each of them.

The mHG test statistic is defined as the smallest of these $P$-values

$$S^{mHG} = \min_t \; p_{(t)}^{HG}$$

In fact, the mHG test applied to gene $G$ tests the null hypothesis that there is no enrichment in cells from cluster $K$ at the top of $S$ against the alternative hypothesis that enrichment exists and that there is a threshold $t^*$ above which enrichment is the most significant. A $P$-value for the mHG test can be computed efficiently using a dynamic programming approach (Eden *et al*, 2007).

The XL-mHG test is a generalization of the mHG test, which introduces two parameters $X$ and $L$ that limit the threshold search space (preprint: Wagner, 2015a). $X$ specifies the minimum number of 1's required to pass the chosen threshold, i.e., it provides a lower bound on the number of true positives. $L$ specifies the lowest cutoff to be examined and provides thus a way to control the maximum number of false positives (this maximum number is $L–X$). Together, these parameters provide a flexible trade-off between the sensitivity and robustness of the test, and prevent COMET from picking up differentially expressed genes that are not relevant as markers due to low specificity or sensitivity. When $X = 0$ and $L = n$, the XL-mHG test reduces to the mHG test.

In COMET, $X$ and $L$ are set to $X = 0.15|K|$ and $L = 2|K|$ by default. When searching for an expression cutoff to binarize gene expression data, we require at least 15 % ($X$) of cells in $K$ to be above the cutoff value and that the number of cells above the cutoff value does not exceed twice the size of $K$ ($L$). We consider the values ($X = 0.15|K|$ and $L = 2|K|$) to be biologically reasonable. We validated on simple examples that they result in desirable properties for the test. In particular, we compared ($X = 0.15|K|$ and $L = 2|K|$) to other values in simulated data including ($X = 0$, $L = n$) corresponding to the basic mHG test (see Fig 3B and Appendix Fig SS4). In these plots, it is shown that the basic mHG test does not enjoy these desirable

properties; hence, $X$ and $L$ need to be picked carefully. While in this manuscript we focus on parameters $X = 0.15|K|$ and $L = 2|K|$ because we believe they provide good predictions of surface marker panels, we emphasize that the user can change the values of $X$ and $L$ when running COMET. The specific $X$ and $L$ values chosen do not affect the performance of the COMET algorithm, but only the quality of markers discovered by COMET. The choice of the ideal $X$ and $L$ parameters for a given marker panel detection problem should be set by the user based on their willingness to tolerate false positives and false negatives.

## Properties of the XL-mHG test

As a rank-based, non-parametric and flexible test, the XL-mHG test has many desirable properties compared to standard differential expression (DE) tests used in single-cell transcriptomics. These properties are shown in Fig 3 and Appendix Figs S4 and S5, and constitute a key asset of the COMET framework. Specifically, in Fig 3B we analyze the performance of the XL-mHG test compared to other differential expression tests (Welch's *t*-test, Wilcoxon rank-sum test, Kolmogorov–Smirnov test, and likelihood ratio test for logistic regression). Data are simulated for one gene in many cells across two clusters: the cluster of interest and another cluster including all the remaining cells (see Simulations subsection below).

Figure 3B-left displays, for each test, the corresponding *P*-value (averaged over several simulation runs) versus the difference in means between the two simulated clusters (the cluster of choice and the background cluster), where the number of cells simulated is fixed. For the classic DE tests, the bigger the mean difference $\varepsilon$, the lower the *P*-value. We observe that the *P*-values for these tests drop below the 0.05 significance level for $\varepsilon$ close to 0.4. Note that in this case, the gene is indeed differentially expressed; however, it would constitute a poor marker for the cluster of interest as its distribution in this cluster overlaps too much with its distribution in the background cluster. The XL-mHG test, however, only picks up significance for $\varepsilon$ close to 4. In that case, the expression distributions in both clusters are much less overlapping, which makes the gene a good marker candidate.

Figure 3B-right displays, for each test, the corresponding *P*-value (averaged over several simulation runs) versus the sample size (number of cells simulated from each cluster, where the cluster of interest represents a fixed percentage of the total number of cells, see Materials and Methods) for a fixed and small value of the difference in means between the cluster of interest and the background cluster ($\varepsilon = 1$). Note that in this case, the gene is effectively differentially expressed across the clusters. For the classic DE tests, *P*-values gradually decrease with sample size, since higher sample sizes correspond to higher effect sizes, and hence a higher statistical power for these tests. While the simulated gene distributions are slightly different between the clusters, due to the small size of this difference, $\varepsilon = 1$, the gene should not be considered a good marker because it would not achieve good specificity or sensitivity. Fortunately, the XL-mHG *P*-values do not drop below significance level for this gene regardless of sample size. This suggests that the XL-mHG test will ignore poor markers regardless of sample size, which is a desirable feature, especially when analyzing scRNA-seq datasets which could be very large.

## COMET algorithm

The COMET algorithm takes as input a gene-by-cell expression matrix (normalized or raw counts), cluster assignments, and visualization coordinates (Appendix Fig S2). Let $X$ denote the full expression matrix (with $n$ cells and $p$ genes), where $X_{ij}$ corresponds to the expression level of gene $i$ in cell $j$. Let $\mathcal{C} = \{C_1, \ldots, C_k\}$ be the set of cell clusters present in the data. COMET will output for each cluster a directory with ranked lists of marker panel candidates consisting of 1, 2, 3, and/or 4 genes, along with informative statistics and visualizations. COMET incorporates several parameters that are set to default values as described below and can alternatively be input by the user. Given a gene-by-cell expression matrix, COMET initially reduces to selected genes (for example, if we are interested in surface markers for live sorting of cells, it is beneficial to use the default list of surface markers available on the COMET web interface). Following this reduction, COMET runs an independent analysis for each of the clusters present in the input file, identifying marker candidate panels for each cluster separately. In each such cluster-centered run, COMET utilizes the XL-mHG test to identify for the given cluster, $K \in \mathcal{C}$ (of size $m$), and each gene, $G$, the optimal threshold and *P*-value. For each gene, a corresponding gene negation is created as a separate gene to be considered by negating the expression of the original gene (multiplying all values by $-1$). This gives us negations as free-standing genes not beholden to their positive counterparts. Any statistical analysis is run on all of these genes, and if a negation is a good marker, it will show up highly in the ranked list based on its individual performance in the panels.

Given the returned thresholds, a binary gene-by-cell expression matrix $A$ is generated, in which $A_{ij} = 1$ if gene $i$'s expression in cell $j$ is above the chosen XL-mHG threshold for gene $i$ (and $A_{ij} = 0$ otherwise). This discretization institutes a "slide-up" policy such that the values for which the matrix element becomes 1 are anything above the chosen expression threshold (e.g., if zero is chosen, we slide the threshold up to just above zero). Note, that since there is an independent run for each cluster, in which cluster-specific XL-mHG thresholds are determined for each gene, the binary matrices generated are cluster-specific and differ across the different parallel runs (one run for each cluster). The binary matrices generated are then used to compute true-positive and false-positive statistics for each gene. Importantly, the binarized matrices are used for assessing gene combinations as described below.

COMET outputs a directory for each cluster. Within each cluster directory, subdirectories are generated, one for each size of combinatorial marker panel computed (e.g., 1-, 2-, 3-, and 4-gene panels, each within a different subdirectory). Within each subdirectory, suggested marker panels are listed in a ranked order, along with informative statistics such as the XL-mHG *P*-value, the XL-mHG threshold, the expression log2 fold change, and the true-positive and true-negative rates.

### Computing and ranking 1-gene marker panels

Single-gene markers are assigned two ranks: the first by their XL-mHG *P*-value (see above) and the second by their absolute log2 fold change of mean expression within and outside of the cluster of interest $K$. A final aggregated rank is assigned to each gene as the arithmetic average of both the XL-mHG *P*-value rank and the absolute log2 fold change rank. We deemed this simple rank aggregation

rule (Boulesteix & Slawski, 2009) relevant and parsimonious given that the two rankings cannot be considered independent from one another. The top-ranking genes are visualized for their continuous expression as well as their binary assignments using the visualization coordinates (e.g., t-SNE or UMAP) provided by the user. All ranks are returned to the user. For the final rankings, a filter is placed on single genes such that any gene with a true-positive rate of < 15% is dropped. Values below 15% for an X value of 0.15 are only possible when the threshold is slid up (as described above) to skip a large number of expression values in the discretization step. Additionally, COMET drops from its ranking negation genes with true-negative rate below 50%. Three output files are then given to the user. One contains the final rankings for all genes, another contains the final ranking for only non-negations (positive markers), and the last contains the statistics calculated for all single genes, unranked and unfiltered to give the user access to all input genes. Each file will also contain various other statistics, such as the *t*-test *P*-value and the Wilcoxon rank-sum test *P*-value along with true-positive-by-true-negative plots of the highest-ranking genes to give the user a broad range of tools to be used in informing their decisions.

### Computing and ranking 2-gene marker panels

The binary expression matrix $A$ is filtered based on the true-positive rate, keeping genes with a true-positive rate equal to or higher than 15%. Let $\tilde{A}$ denote the restriction of matrix $A$ to such genes, and $\tilde{A}_{|K}$ denote the restriction of matrix $\tilde{A}$ to cells in cluster $K$. We further define the matrices $R = \tilde{A}\tilde{A}^T$ and $R_{|K} = \tilde{A}_{|K}\tilde{A}_{|K}^T$. We call $R$ the matrix of positives, and $R_{|K}$ the matrix of positives in cluster $K$. The *i,j*-th element of $R$ (resp. $R_{|K}$) is the number of cells (resp. cells in cluster $K$) in which genes $i$ and $j$ are co-expressed. This procedure only relies on the binarization obtained via the XL-mHG test for single genes. No parametric form is assumed for the relationship between the cell cluster labels and the binarized expression of single genes and their combination (this statement holds for combinations of an arbitrary number of genes).

Using the matrices $R$ and $R_{|K}$ (along with the cluster size $m$), a hypergeometric enrichment *P*-value can be computed for each 2-gene combination. Gene marker panels are ranked by the hypergeometric enrichment *P*-value as well as a "Cluster Clear Score" (CCS) which evaluates the extent to which a panel cleans out contaminating clusters, as compared to using a single gene from the combination. The CCS value for a given gene pair is as follows:

$$CCS = \sum_{C \in \mathcal{C} \setminus \{K\}} TN_C^{after} - TN_C^{before}$$

where $TN_C^{before}$ is the true-negative percent in cluster $C$ for the single gene in the pair with the lowest *P*-value (the "lead" gene) and $TN_C^{after}$ is the true-negative percent in cluster $C$ for the pair (after addition of the second gene). The final COMET score given to 2-gene combinations is a simple average between the hypergeometric rank (in ascending *P*-value order) and the CCS rank. Only the top 1,000 hypergeometric test-ranked pairs are considered in the CCS. The CCS favors gene combinations that "clean out" specific contaminating clusters in an attempt to assist in finding solutions for clearing out entire contaminating populations with the marker panel. If other measures of ranking are preferred, users can use the global TP and

TN rates reported for each gene panel to construct alternative statistics to rank by. All gene negations are treated as stand-alone genes and will be fully incorporated in the pair output. Following ranking, plots are generated for the top-ranking pairs, showing the binary expression of each of the paired genes and their combination, for efficient parsing of the results by the user (as shown in Fig 6). Each "lead" gene is limited to appearing in at most 10 plots to avoid "takeover" by a gene that ranks highly multiple times due to pairing with many equivalent genes. Any one gene is allowed to appear at most 200 times in the output folder.

### Computing and ranking 3-gene marker panels

The construction of the matrices of positives $R$ and positives $R_{|K}$ in cluster $K$ in the 3-gene case is fairly similar in process to the pair expression matrix. We opted for a matrix representation of the 3-gene case for reasons of computational speed and memory. If we consider the binarized expression matrix $\tilde{A}$ (where $\tilde{n}$ denotes the number of cells and $\tilde{p}$ denotes the number of genes), we can construct a $\tilde{p}^2$-by-$\tilde{n}$ matrix $Q$ that includes the count expression for each gene pair within a given cell by performing a row-wise AND operation on two copies of the original $\tilde{A}$ (for each two rows in original matrix $\tilde{A}$). Then, $Q\tilde{A}^T$ is a $\tilde{p}^2$-by-$\tilde{p}$ matrix where each entry represents the number of positive cells for a three-gene combination. Two of these matrices are constructed, one for all cells within the cluster $K$ and one for all cells. This gives us everything necessary to compute a hypergeometric *P*-value, true-positive rate, and true-negative rate. Once each combination has these statistics computed, the tool will filter out combinations in the output file that are either redundant or useless. For example, any gene combination that is repeated (e.g., combination BAC if combination ABC already exists) can be removed or any combination with repeating genes (e.g., combination AAB) can also be removed. With these types of panels removed, the resulting data consist only of unique 3-gene marker panels. They are ranked by their hypergeometric *P*-value. These combinations are only available on the software package, along with an option to compute the 3-gene combinations via a heuristic to enable significant speed-up in cases where the number of genes and/or cells is large.

### Computing and ranking 4-gene marker panels

For four genes, generating the matrices of counts for positives and positives-in-cluster is similar to the three- and two-gene cases. We consider the upper triangle of values of $QQ^T$, where $Q$ is as defined above. Duplicates and gene-repeating combinations are once again filtered out, and the resulting entries contain unique 4 gene marker panels. The 4 gene combinations are ranked by the *P*-value of the hypergeometric test, with the true-positive and true-negative rates computed and placed in the results table. The top-ranking thirty combinations are plotted using the visualization coordinates supplied by the user. Four-gene combinations are currently available only in the software package.

### Automation and deployment of COMET

In order to enable easy access to the COMET, tool we set up an automated web interface in addition to the COMET software package. All of the COMET software is freely available. The COMET web interface is deployed onto an Amazon Web Services (AWS) platform with the back end fully controlled by a Flask (Python) application

(Appendix Fig S11). A front-facing interface environment pulls in the user's files and dispenses them to the automation line. Each job submitted is sent to a "Simple Queue" which stacks jobs on a first-come, first-serve basis. Demo jobs are processed in a separate but identical pipeline to ensure they do not take priority over real jobs. A working environment on a different AWS instance polls the queue for jobs, processing one at a time. Upon completion of a job, the worker will send out an email to the user with a job ID (a thirty-two-character UUID) which is then entered by the user on the COMET website under the "Results" section to enable downloading of the results. Files are stored in an "S3" bucket, which can store large files, and are deleted after 4 days. Files can be accessed by any user with the proper job ID so that sharing of results is simple.

### Heuristic analysis

To reduce time of computation, we implemented heuristics for the generation of ranked 2-gene, 3-gene, and 4-gene combinations. The heuristic algorithm can be opted for use by the user in cases of large gene or cell counts. In the heuristic approach, we chose a heuristic core size, $N$, and compute 2-gene or 3-gene combinations. This core size tells us the number of top-performing single genes we will use to pair with all other genes, creating a new search space. In the 2-gene case, we take the top $N$ genes (based on their XL-mHG $P$-value performance) and pair with all other genes. In the 3-gene case, we first mimic the 2-gene case above by finding all pairs between the top $N$ genes and all other genes, and then match those up with genes from the same list of $N$ genes to create the triplet. Varying the threshold value $N$ allows us to see which size of heuristic performed best in comparison with running the full, non-heuristic search space of gene combinations (see results from the splenic follicular B-cell cluster, as shown in Fig 5A, analyzed in Fig 2C and D). For 4-gene combinations, the heuristic is to take the top $N$ genes and only create combinations using genes in that list. In general, there is a certain threshold level for each dataset that will yield the closest to fully correct results while significantly speeding up the time of computation. By default, COMET currently uses a heuristic core size of 50. There are also heuristic core sizes for which the heuristic performs poorly, namely at 5 (Fig 2D). Since the CCS ranking is only computed for the first 1,000 gene pairs, there is a significant shuffling of the expected and heuristic ranks at these low levels; thus, it is necessary for users to stay at suitably high choice of heuristic core size to maintain accurate results.

### Simulations

#### Comparison of COMET to standard differential expression tests

All simulations are conducted in Python 3.6 using the Pandas, Sklearn, NumPy, and SciPy core functions as well as the xlmhg package (Eden et al, 2007; Wagner, 2015b). We consider a gene $G$ expressed in a population of $n$ cells comprising two clusters $K$ and $C$, where $K$ denotes the cluster of interest ($p$ cells). In practice, we would have $C = K^c$, where $K^c$ denotes the complement set of $K$. Let $X_1, X_2, \ldots, X_m$ (resp. $X_{m+1}, X_{m+2}, \ldots, X_n$) be the random variables corresponding to $G$'s expression level in $K$ (resp. $C$). In these simulations, we assume that $\{X_i\}_{1 \leq i \leq m}$ (resp. $\{X_i\}_{m+1 \leq i \leq n}$) are mutually independent and identically distributed, and that the $\{X_i\}_{1 \leq i \leq m}$ and the $\{X_i\}_{m+1 \leq i \leq n}$ are independent. We further denote by $C_i \in \{0, 1\}$

the variable indicating cluster membership of the $i$th cell, $i = 1, \ldots, n$, where 1 refers to cluster $K$ and 0 refers to cluster $C$.

We compare the XL-mHG test (with default COMET parameters $X = 0.15|K|$ and $L = 2|K|$, where $|K|$ denote the cardinal of set $K$) to standard statistical tests that may be used in single-cell RNA sequencing for gene differential expression (DE) analysis, including Welch's $t$-test (scipy.stats.ttest_ind with equal_var=FALSE), the Wilcoxon rank-sum (WRS) test (scipy.stats.ranksums), and the Kolmogorov–Smirnov (KS) test (scipy.stats.ks_2samp). We also consider the likelihood ratio test (LRT) for the logistic regression model (Ntranos et al, 2019)

$$C_i|X_i \sim Bernoulli(\sigma(\beta_0 + \beta_1 X_i))$$

where $\sigma(\cdot)$ is the logistic function defined as $\sigma(t) = 1/(1 + e^{-t})$. The null model for the LRT is that cell membership to a cluster does not depend on gene expression, i.e., $\beta_1 = 0$. Logistic regressions are performed using sklearn.linear_model.LogisticRegression.

We simulate gene expression values from a Gaussian model (using numpy.random.normal) in Fig 3 and Appendix Fig S4. In cluster $K$, we let $\{X_i\}_{1 \leq i \leq m}$ be independent Gaussian draws with mean $\varepsilon$ and standard deviation 1, and in cluster $C$, we let $\{X_i\}_{m+1 \leq i \leq n}$ be independent Gaussian draws with mean 0 and standard deviation 5, where $P = 0.1n$. Gene $G$ is differentially expressed as soon as $\varepsilon \neq 0$, yet it is considered a good marker when $\varepsilon$ is high enough for the gene expression distributions in $K$ and $C$ to be well separated (for example, when $\varepsilon \geq 3$). Results are averaged over 100 simulation runs, and reported error bars on these $P$-values correspond to one empirical standard deviation over the 100 runs (Appendix Fig S4).

We also simulate gene count data from a negative binomial model (using numpy.random.negative_binomial) in Appendix Fig S4. In cluster $C$, we let counts follow a negative binomial distribution $NB(m = 1, \pi = 0.1)$ where $m$ denotes the number of successes (extended to the reals) and $\pi$ denotes the probability of success in the standard negative binomial representation. For cells in $K$, gene count values follow a $NB(m = 5, \pi = 0.5)$ distribution shifted by $(4 + \varepsilon)$, so that the difference in means between the two clusters is precisely $\varepsilon$. We still have $P = 0.1n$, and results are averaged over 100 simulation runs (Appendix Fig S4).

#### Comparison of COMET to classifiers

We compare the XL-mHG procedure (with default COMET parameters $X = 0.15 * |K|$ and $L = 2|K|$) to standard classifiers used in single-cell transcriptomics for gene DE analysis or other related tasks, including random forests and logistic regression. Recent literature mentions the use of LR in gene DE analysis (Ntranos et al, 2019), while RF have been used for a variety of tasks in genomics (Irrthum et al, 2010).

#### Gaussian generative model

We first resort to a simple Gaussian simulation engine. We simulate an $n \times p$ cell-by-gene Gaussian expression matrix (see Appendix Fig S5), where $n = 500$ cells and $p = 1,000$ genes. Cells belong either to the cluster of interest $K$ (10% of cells) or the super-cluster $C$ (90% of cells). The gene structure is threefold: (i) $x\%$ of the genes are not $K$ markers ($G_1, \ldots, G_r$ have Gaussian expression with mean 0 and

standard deviation 1 in both clusters), (ii) 5% of the genes are good $K$ markers ($H_1$, …, $H_s$ have Gaussian expression with mean 1 and standard deviation 1 in cluster $C$ and mean 1 and standard deviation 1 in cluster $K$), and (iii) $z$% of the genes are poor $K$ markers ($I_1$, …, $I_t$ have Gaussian expression with mean expression $\mu_p$ and standard deviation 1 in a random subset of $w$% of cells in $K$ and with mean expression 0 and standard deviation 1 in all other cells). Note that $z = (95-x)$. Poor markers can biologically correspond to measurement outliers or markers for subclusters of the population of interest (Appendix Fig S5).

The XL-mHG procedure, logistic regression, random forest, and an extra trees classifier are applied to the dataset in order to rank the genes in terms of their potential as markers for cluster $K$. Ranking for the XL-mHG procedure is obtained using the XL-mHG *P*-value associated with each gene. For logistic regression, we use the likelihood ratio test *P*-value using the same null model as described above. To ensure comparability across simulation runs, random forest parameters were set to the following standard values. The random forest model (sklearn.Ensemble.RandomForestClassifier) was trained using $N_T = 250$ trees. Each tree uses a size-$n$ bootstrap sample from the original dataset, and a subset of genes of size $\sqrt{p}$ was used to split at each node. Trees were grown until all leaves achieve Gini purity. Gene ranking utilizes the Gini importance metric (see below). The same parameters were used to train an extra trees classifier (sklearn.Ensemble.ExtraTreesClassifier). An extra trees classifier differs from a random forest classifier in that candidate split points are drawn uniformly at random from the range of each selected feature at a given node, as opposed to deterministically chosen in random forest.

A random forest is an ensemble of classification trees built using a learning dataset of size $n$ where each cell is labeled and characterized by $p$ genes. A node $t$ defines a partition of the sample space based on the binary test of one gene (or split, $s_t$), selected from a random subset of all genes. In each tree, a recursive learning procedure identifies at a given node $t$ the split $s_t$ for which the partition of the $n_t$ node samples into $t_{left}$ and $t_{right}$ maximizes the decrease

$$\Delta i(s, t) = i(t) - p_{left} i(t_{left}) - p_{right} i(t_{right})$$

of the Gini impurity measure $i(\cdot)$, and where $p_{left} = N_{t_{left}}/N_t$ and $p_{right} = N_{t_{right}}/N_t$.

It is then possible to evaluate the importance of gene $G$ for predicting cluster label by adding up the weighted impurity decreases $p(t)\Delta i(s_t,t)$ for all nodes $t$ where $G$ is used, averaged over all $N_T$ trees in the forest:

$$MDG(G) = \frac{1}{N_T} \sum_T \sum_{t \in T: v(s_t) = G} p(t)\Delta i(s_t, t)$$

where $p(t)$ is the proportion of samples reaching node $t$ and $v(s_t)$ is the gene used in split $s_t$. We utilized this metric, christened Gini Importance (or Mean Decrease Gini, MDG) (Louppe *et al*, 2013), to rank the genes in our random forest simulations.

We define the scaled sum of ranks (SSR) as a metric to quantify the performance of XL-mHG, RF, and LR at recovering good markers from simulated data matrices. Note that genes $H_1$, …, $H_s$ are all equally good markers, with no intrinsic ranking among them. For example, if $S_s$ denotes the symmetric group of $\{1,\ldots,s\}$, then for

any permutation $\pi \in S_s$, the rankings $r_0 = (H_1,\ldots,H_{s-1},H_s)$ and $r_\pi = \left(H_{\pi(1)},\ldots,H_{\pi(s-1)},H_{\pi(s)}\right)$ are exactly equivalent. The SSR metric is designed to respect this property (in the example, it gives an equal score to both rankings $r_0$ and $r_\pi$). We compute the rank assigned to each of $H_1$, …, $H_s$ by a given method (XL-mHG, RF, or LR), and then sum up these ranks and divide by the optimal sum of ranks $s(s+1)/2$. In other words,

$$SSR(M) = \frac{2}{s(s+1)} \sum_{j=1}^{s} rank(j|M)$$

where $M$ refers to the method used to rank the genes (XL-mHG, RF, or LR). A good marker discovery procedure will recover most of the markers and result in SSR values very close to 1, while a suboptimal marker discovery procedure will result in higher SSR values. Results in Fig 3C and Appendix Fig S5 are averaged over 20 simulation runs.

In Fig 3C and Appendix Fig S5B, poor markers constitute 10% of the total number of genes and are expressed in 10% of cells in the cluster of interest $K$. In Appendix Fig S5C, poor markers are expressed in 10% of cells in $K$ with a mean expression value $\mu_p = 30$. The XL-mHG procedure outperforms RF and LR in detecting good markers from the simulated gene expression matrix. While poor markers are suboptimal for the purpose of correctly classifying the cells, logistic regression will detect these genes if they induce a high fold change between the cluster of interest $K$ and the remaining cells in $C$. RF will leverage poor markers as they can marginally improve classification while still being expressed in $< 15$% of cells in $K$.

### Noisy Poisson–Gamma generative model

We confirm these results using a more complex generative model for single-cell RNA-seq data, so as to capture the noise inherent to transcriptomic datasets. UMI-based single-cell sequencing is subject to two main types of technical (i.e., non-biological) variability: efficiency noise and sampling noise (Grün *et al*, 2014; preprint: Wagner *et al*, 2019). The efficiency noise captures the global cell-to-cell variation in sequencing efficiency. The sampling noise is related to low detection rates of mRNA molecules in each cell and is well approximated using Poisson resampling. In the same spirit as in the previous simulation engine, we simulate an $n \times p$ cell-by-gene count matrix (see Appendix Fig S6A, right), where $n = 500$ cells and $p = 1,000$ genes. Cells belong either to the cluster of interest $K$ (10% of cells) or to the super-cluster $C$ (90% of cells). The gene structure is threefold: (i) $x$% of the genes are not $K$ markers (this set of genes is called $\mathcal{N}$), (ii) 5% of the genes are good $K$ markers (this set of genes is called $\mathcal{G}$), and (iii) $z$% of the genes are poor $K$ markers (this set of genes is called $\mathcal{P}$, and for each gene $G \in \mathcal{P}$, $G$ is expressed in a random subset of $q$% of cells in $K$, denoted as $K_{|G}$). Note that $z = (95-x)$.

The first step of our simulation procedure consists in generating a cell-by-gene matrix of true counts (ground truth) using a hierarchical Poisson–Gamma model (Appendix Fig S6A, left). For each gene $g$ belonging to a particular gene type $T$ (where $T \in \{\mathcal{N}, \mathcal{G}, \mathcal{P}\}$) in a given cell population $P$ (where $P \in \{K, C, K_{|g}\}$), the mean expression $\mu_g^P$ of gene $g$ is drawn from a $Gamma(\alpha_{TP}, \beta_{TP})$ distribution. The true transcript count $X_{gi}$ for gene $g \in T$ in cell $i \in P$ is then drawn from a $Poisson(\mu_g^P)$. The matrix $X = \left(X_{gi}\right)_{g,i}$ corresponds to the ground truth

on top of which we will subsequently add technical noise (Appendix Fig S6A, center). For each cell $i \in K \cup C$, an efficiency scaling factor $s_i$ is drawn from a $Uniform([1-e, 1+e])$ distribution, where $e \in (0,1)$ is a parameter. For a given gene $g$, the noisy version of $X_{gi}$ is obtained by resampling $Z_{gi} \sim Poisson(s_i X_{gi})$. The matrix $(Z_{gi})_{g,i}$ represents the observed (noisy) cell-by-gene count matrix that we use in our simulations to compare the XL-mHG procedure to standard classifiers including RF, XT, and LR.

Simulations utilize the following parametrization (note that $\alpha_{\mathcal{P}K_{|g}}$ and $z$ will vary across a range of values in Appendix Fig S6). For every gene type $T$ and cell population $P$, we set $\beta_{TP} = 0.1$. We set $\alpha_{\mathcal{G}K} = 2^5$, $\alpha_{\mathcal{G}C} = 2^4$, $\alpha_{\mathcal{N}K} = \alpha_{\mathcal{N}C} = 2^4$. For a given poor marker gene $g \in P$, we set $\alpha_{\mathcal{P}K_{|g}} = 2^{10}, \alpha_{\mathcal{P}K_{|g}^c} = 2^4$ where $K_{|g}^c$ refers to the complement cell set of $K_{|g}$. We further set $q = 10$, $e = 0.2$, and $z = 10$. Results are provided in Appendix Fig S6B and C and are similar to the ones obtained using our simpler Gaussian simulation engine. In particular, the XL-mHG procedure outperforms other classifiers in recovering good markers from the noisy scRNA-seq count matrices (low SSR). This does not prevent RF and XT from being good classifiers as shown by the low out-of-bag error achieved by both tree ensemble methods.

### Comparison of single-marker predictions for splenic cell populations

To compare marker detection approaches, we used the Mouse Cell Atlas spleen population (Han et al, 2018). Expression data, cluster assignments, and t-SNE coordinates were downloaded from the MCA Gallery (MCA—Mouse Cell Atlas). The gene-by-cell count matrix was normalized by dividing each count by the sum of counts in the cell and multiplying by the median gene count across cells. Methods used for comparison to COMET included the t-test (SciPy), Wilcoxon rank-sum test (SciPy), likelihood ratio test with a logistic regression model (Sklearn, Python), MAST (R), and the XL-mHG test. COMET was run using the cell surface gene list available on the COMET web interface (Chihara et al, 2018). Each test besides MAST was integrated into the statistics generated by COMET to give the rankings for each test in a single data file, while MAST was run in a separate R script. To choose marker genes, we used recommendations from an established table (BioLegend Essential Markers for Phenotyping) to construct a short list of genes for each cell type subpopulation. The subpopulations of monocytes, granulocytes, and neutrophils showed poor marker results in all of the methods (potentially due to small cluster sizes) and were removed from the analysis.

### Computation of gene signatures

Scores for gene signatures were calculated using Scanpy's *score_genes* method (Wolf et al, 2018), and code is available on our GitHub (https://github.com/MSingerLab/COMETSC).

### Identification of marker panels for splenic follicular B cells

B-cell subpopulation markers were determined using COMET on the Tabula Muris (Tabula Muris Consortium, 2018) mouse spleen dataset and the cell surface gene list available on the COMET web interface (Chihara et al, 2018). Expression data, cluster assignments, and t-SNE coordinates were downloaded from the Tabula Muris website and trimmed to only include cells from the B-cell clusters. COMET was run with its default parameters of X and L.

### Experimental materials and methods

#### Mice

6- to 8-week-old, female, wild-type C57BL/6J mice were purchased from the Jackson Laboratory. Mice were housed under SPF conditions. All experiments involving the use of laboratory animals were approved by and carried out in accordance with the guidelines of the Brigham and Women's Hospital (BWH) Institutional Animal Care and Use Committee (IACUC) (Boston, MA).

#### Immune cell isolation

1   Spleens were isolated from wild-type C57BL/6J mice.
2   Spleens were smashed using a 40-μm sterile filter (Fisher Scientific) and spun down at 350 g, 4°C for 5 min.
3   Pellets were resuspended in 1 ml ACK lysis (Buffer from Gibco) and incubated for 5 min at room temperature.
4   Cells were washed with 10 ml flow buffer (DPBS with 2% FCS) and spun down at 350 g, 4°C for 5 min.

#### Flow cytometry

1   Single-cell suspensions were stained in 1 ml flow buffer (DPBS with 2% FCS) at 4°C for 30 min with antibodies against cell surface proteins in the concentrations listed in the Reagents and Tools table.
2   Cells were washed with 10 ml flow buffer (DPBS with 2% FCS) and spun down at 350 g, 4°C for 5 min.
3   Cells were ready for analysis on the flow cytometer. All samples were run on a BD LSR II (BD Biosciences) or LSRFortessa (BD Biosciences).

The flow cytometry data were analyzed with FlowJo software (Tree Star) and Prism (GraphPad). The cells were gated for single and viable lymphocytes and then analyzed for the respective expression of the surface molecules of interest.

# Data availability

The computer code for COMET is available at https://github.com/MSingerLab/COMETSC. FACS data generated in this study were deposited in https://flowrepository.org/.

**Expanded View** for this article is available online.

## Acknowledgements

We thank Florian Wagner, Zohar Yakhini, Lloyd Bod, and Brian Cleary for insightful discussions and the reviewers of this manuscript for their constructive comments. We thank Christina Usher for help with illustrations. AS was supported by a German Academic Scholarship Foundation (Studienstiftung des Deutschen Volkes) PhD fellowship. This manuscript was supported in part by the Parker Institute for Cancer Immunotherapy. This study was supported in part by NIAID of the National Institutes of Health under award number P01AI129880.

## Author contributions

All authors contributed extensively to the design of the COMET tool and interpretation of the results. MS and AS conceived the study. CD and AY-S designed and developed the software tool and documentation. AS designed and performed mouse experiments. LVC designed and performed all statistical analyses and supported the development of the software tool. CD, AS, LVC, AR, VKK, and MS wrote the manuscript, and AR, VKK, and MS supervised the project.

## Conflict of interest

The authors declare that they have no conflict of interest.

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
