## [Review Process File · Molecular Systems Biology]

Combinatorial prediction of marker panels from single-cell transcriptomic data

Conor Delaney, Alexandra Schnell, Louis V. Cammarata, Aaron Yao-Smith, Aviv Regev, Vijay K. Kuchroo and Meromit Singer.

Review timeline:

Submission date:	26 th May 2019
Editorial Decision:	14 th July 2019
Revision received:	8 th August 2019
Editorial Decision:	6 th September 2019
Revision received:	18 th September 2019
Accepted:	20 th September 2019

Editor: Maria Polychronidou

Transaction Report:

1st Editorial Decision

14th July 2019

Thank you again for submitting your work to Molecular Systems Biology. We have now heard back from the three referees who agreed to evaluate your study. As you will see below, the reviewers acknowledge that the addressed topic is relevant and think that Comet could potentially be a useful tool for analyzing scRNA-seq data. They raise however a series of concerns, which we would ask you to address in a major revision.

All three reviewers provide clear and constructive recommendations on how to improve the study and there is therefore no need to repeat the comments listed below. Reviewer #3 expresses concerns regarding the novelty of the method and its advantages compared to existing approaches, and we would ask you to clarify these points. Please feel free to contact me in case you would like to discuss in further detail any of the issues raised by the reviewers.

REFeree REPORTS

Reviewer #1:

This paper proposes a combinatorial approach to find gene-marker panels (of size 1 to 4 genes) for single-cell RNAseq data. It ranks the candidate marker panels with respect to their resulting p-value, log-fold change, and true positive and true negative rates. The authors show, for splenic and splenic follicular B cells, that the top ranking genes of their tool are favourable for flow-cytometry staining of B cells. Overall, the the idea of the paper is very interesting and the addressed problem is an important one in this field. I ask here for a number of clarifications and extensions:

1- "Genes are also tested for their potentials to be used as negative markers". It is not clearly

explained in the main text (also in Methods) how negative markers are identified by COMET.

2- "We set the default parameters for X and L within COMET to be 15 of the size of the cluster of interest and twice the size of the cluster of interest." It has not been validated in the paper whether X and L are properly set to 15% and 50%. For example, a comparison with the basic mHG technique would be interesting.

3- "we chose a heuristic core size, N". What does a core size mean in this concept?

4- It is very surprising that random forest does not work well here. I would suggest to report the error rate of the random forest (out-of-bag error) as well to ensure that your implementation was a good one.

5- Page 27: SSR sums up the rank of 's' genes and divides that by the optimal sum of ranks to quantify the performance of different classifier techniques. Why don't you measure the absolute difference between the predicted and the optimal ranks?

6- "A final rank is assigned to each gene as the average of both XL-mhG p-value rank and the absolute log₂-fold-change rank." It is not very clear how this ranking approach works. However, a well-know policy in the optimization domain is that the objectives (here p-value and log₂-fold-change) are normalized first and the weighted sum or subtract of them are used for ranking.

7- Page 38, "with respect to both robustness to small effect-size". What does "effect size" mean in this concept?

8- Fig. 3 (B): An explanation on how to interpret these plots will help a lot to understand them.

9- The authors compared their proposed approach with the related work only using synthetic data. It is interesting to see if COMET still outperforms other approaches for real data as well. For example, in figures 4 and 5, what are the gene-marker panels that previous approaches find?

10- In Fig. 5 (D), why CXCR4 is performing better while being "rank 3"?

11- Figure S2 (A): How the user should select among the top-ranked panels?

Finally, some minor issues:

1- The quality of Figure s6 is not high enough.

2- The legend in Figure S8 (C) is not readable.

Reviewer #2:

Here, Delaney et al present Comet, a method for combinatorial prediction of marker genes. Having clustered cells from scRNA-seq experiments, one of the most important questions is often to identify genes that can be used to uniquely identify each cluster. Many clustering methods provide such a feature, but the selection is typically carried out by investigating one gene at a time. It is clear, at least from a theoretical point of view, that this may lead to sub-optimal solutions. Thus, Comet could potentially be an important tool for analyzing scRNA-seq data.

The current manuscript is well-written with very few typos and easy to follow. Nevertheless, I have the following major issues:

I had problems installing the software (I emailed about this). Unfortunately, their fix did not work for me as I only have python 3.7 installed and not 3.6. I tried a few different fixes (I am not an expert on python nor on virtual environments) but I was unable to get it to work. Although there is an obvious solution to this issue (which I did not try), I do think that it highlights a potential issue,

as time goes by fewer people will have python 3.6 installed and most people would probably prefer if they did not have to install it just to run comet. If it is possible to run with higher versions of python, then I suggest that you add instructions to your installation guide. If not, it would be a good idea to state this requirement up front more explicitly than it is now.

I then tried the web interface with the Tabula Muris 10X heart dataset (624 cells). After some struggles with the formatting, I got the job to run, but when I downloaded the results, all the files were empty and there was no error message. This job was ID 0bf44897-a554-403a-80d0-92411d8af4ea. It is obviously hard for me to determine what went wrong, but I would recommend that some sort of error message is communicated to the user if this happens.

The authors use several synthetic examples to validate their method. Due to the lack of ground truth, this is an acceptable approach, but I do not think that the authors' have used best practice. Instead of using their relatively straightforward simulation strategy, I would recommend that they instead employ Splatter (<https://genomebiology.biomedcentral.com/articles/10.1186/s13059-017-1305-0>) or some other software which can provide more realistic simulations of scRNA-seq data. The methods employed by the authors are unlikely to capture many of the more complex features of real data. Currently, results are only reported for immune related datasets. However, I would like to see results for other tissues as well (e.g. from Tabula Muris or the mouse cell atlas) to demonstrate that the method generalizes well.

It would also be useful if the authors could comment on the importance of sequencing depth for the performance of the method in addition to the sample size. In experimental design there is often a trade-off between total number of cells and reads/cell, so it is helpful to know what combination is best for marker gene discovery using comet.

Why is it necessary to provide t-sne coordinates as input? I assume that they are used for visualizing the results? If one is only interested in the best genes then this is probably not needed, right? Would it be possible to make this input optional?

In the supplement, the algorithm is presented using matrix algebra terminology. For the 3- and 4-gene cases I was wondering if this could be more elegantly represented using 3- or 4-way tensors rather than the current representation?

The optimization is carried out with respect to the clear cluster score which is based on true negative calculations. Although this is a perfectly reasonable objective function, one could also think of other choices. In particular, it might be useful to instead consider precision or recall. Would it be possible to use other quantities instead of the TNs for the optimization and do they present any advantages?

Reviewer #3:

- This paper contributes another approach to perform differential expression analysis to identify gene sets characteristic for single-cell RNA seq cell type clusters. The authors claim that their work goes beyond mere differential expression analysis and is tailored to panel prediction for functional assays/follow up of discoveries from sc RNA.

- The novelty and significance of this work is limited for the following reasons.

- The differential expression task is not novel, has been defined since the introduction of single-cell RNA sequencing datasets and a plethora of statistical/computational solutions proposed. For instance see Luecken and Theis, 2019 for an overview of best practices on this topic.

- The authors claim there is a lack of panel prediction approaches. In fact, there are quite a few panel prediction approaches out there already. The majority of exploratory sc RNA sequencing studies that claim identification of novel cell subsets, follow up by defining a panel of FACS compatible markers to validate the novel subsets and describe their function. These approaches are conceptually the same as the proposed approach here: identify differentially expressed genes that characterize the cell cluster of interest and use the most prominent ones for flow cytometry based validation. See e.g. Blecher-Gonen et al. 2019, Jaitin et al. 2019, or Villani et al. 2017 (the latter from the Regev lab).

- The study by Delaney et al. demonstrates its differential expression approach by solving a rather simple task, i.e. by recovering a priori known and abundant B cell subsets. Above example studies by Blecher-Gonen et al. 2019, Jaitin et al. 2019, or Villani et al. 2017 make a significantly stronger

point by identifying novel and less abundant cell subsets and by functionally validating these in follow up experiments.

- The proposed approach is conceptually one more agnostic differential expression approach that seems not to be tailored for panel prediction since it does not incorporate any specific cue that enriches for markers that are compatible with flow cytometry based validation.

- The only modest novelty of this work is the incremental methodological contribution of modeling differential expression as an urn experiment, where the binarized gene expression levels follow a hypergeometric distribution under the null hypothesis.

Other Major concerns

- The hyperparameters of the proposed approach seem to be set ad hoc. It would be important to understand the impact of these settings. E.g. what is the sensitivity of the reported results on the threshold definition for gene expression binarization? What is the sensitivity to the X/L settings of the XL-mHG test.

- The authors have separate approaches for evaluating the discriminative power of gene sets of varying size (2-4 tuples). This seems cumbersome. Why do the authors not evaluate regularized classifiers with that efficiently model - possibly a lot larger - gene groups as interaction terms?

- The authors use simulated sc RNA seq data for initial benchmarking. The simulation models seem simple. Why did the authors not resort to publicly available sc RNA seq simulation tools (e.g. Splatter by Zappia et al. 2017)

- It seems that the classifiers are not tuned to avoid overfitting, e.g. by cross validation, i.e. fitting on training data, hyperparameter/model complexity tuning via evaluation on validation data and performance evaluation on test data. This setup allows for identifying suitable RF parameters (and not arbitrarily choosing one parameter set as done currently) and also to identify a regularization strength for the logistic regression (so far logistic regression is used without regularizing the complexity of the fitted model. This is frequently done e.g. by adding an L1 penalty. This lack of proper tuning could explain the result for the classifiers in Fig. 3C, where with sufficiently large effect sizes of the poor markers these are picked by overly complex/overfitted classifiers.

- Fig 3B. This result seems not to be favorable for the XL-mHG test. This test requires much larger effect sizes to achieve a significant DE result and does not achieve significance for any of the benchmarked sample sizes.

- Fig 3D. comparison ranking from for Random Forest missing (e.g. according to feature relevance)

- Fig 4: on what comparison is the COMET result based? XL-mHG Test for genes contrasting B cells and non-B cells? Are all non-B cell types annotated in the HCA atlas merged into one group of cells for this comparison?

- Fig4: what is the ranking produced by the other DE and classification approaches compared to in Fig. 3?

- Fig 4B: what data is shown? The caption indicates that splenic B cells from the MCA data is shown. Why are not all B cells positive for the canonical B cell marker CD 19?

- Figs 4B & C: what besides the colors is different on the two subpanels of Fig 4B and Fig4C? Please indicate in the caption.

Minor issues:

- The authors state that COMET assumes a non-parametric model. That is not correct. The gene

expression binarization is part of the COMET analysis and assumes a parametric model, rendering the whole approach parametric.

- The authors qualify markers identified by their approach as "favorable". This qualification of seems not adequate in the respective contexts. "cell type specific" or some other similar qualifier seem to be more adequate here.

- Fig 3D what is the difference between XL-mHG and COMET. Both not the same?

- What implementations were used for the classifiers?

Response to Reviewer #1:

Reviewer #1: This paper proposes a combinatorial approach to find gene-marker panels (of size 1 to 4 genes) for single-cell RNA-seq data. It ranks the candidate marker panels with respect to their resulting p-value, log-fold change, and true positive and true negative rates. The authors show, for splenic and splenic follicular B cells, that the top-ranking genes of their tool are favorable for flow-cytometry staining of B cells. Overall, the idea of the paper is very interesting and the addressed problem is an important one in this field. I ask here for a number of clarifications and extensions:

A. Major Issues

1- "Genes are also tested for their potentials to be used as negative markers". It is not clearly explained in the main text (also in Methods) how negative markers are identified by COMET.

We thank the reviewer for this comment which highlights the need for clarification. COMET assesses genes for their capability of being negative markers by duplicating each gene, to construct "gene" and "gene_negation", where in "gene_negation" all expression values are multiplied by (-1). Each negation is treated as a standalone gene such that all statistics are calculated irrespective of their positive counterpart. Therefore, any well performing negations will be ranked highly in our panels in the same way a normal gene would. We have clarified this in the text in the Methods section, page 20. The underlined text below was added:

For each gene, a corresponding gene negation is created as a separate gene to be considered by negating the expression of the original gene (multiplying all values by -1). This gives us negations as free-standing genes not beholden to their positive counterparts. Any statistical analysis is run on all of these genes and if a negation is a good marker, it will show up highly in the ranked list based on its individual performance in the panels.

2- "We set the default parameters for X and L within COMET to be 15 of the size of the cluster of interest and twice the size of the cluster of interest." It has not been validated in the paper whether X and L are properly set to 15% and 50%. For example, a comparison with the basic mHG technique would be interesting.

We agree with this reviewer that a more extensive discussion of the values to which the X and L parameters were set to, as well as a discussion of the effect of varying the X and L parameters

and the performance of the basic mHG test, constitute important aspects to be discussed in the manuscript. We discuss each of these subjects below:

When assessing the performance of the basic mHG test, we observe that as expected, the parameters X and L have a significant influence on the performance of the test. The X and L parameters put bounds on the false-negative (X) and false-positive ($L-X$) counts. The parameters ensure that each gene considered as a marker would provide at least X true-positive calls and at most $(L-X)$ false-positive calls. Hence, the parameters X and L reflect the tradeoff between true and false positive rates in the binarization performed by the mHG test. These parameters prevent COMET from picking up differentially expressed genes that are not relevant as markers due to low specificity or sensitivity.

To clarify the above point, we added the basic mHG test performance to Figure 3b and Appendix Figure S4 (see below). Additionally, we added the following sentence to the XL-mHG test section of the Methods section (page 18):

“Together these parameters provide a flexible trade-off between the sensitivity and robustness of the test, and prevent COMET from picking up differentially expressed genes that are not relevant as markers due to low specificity or sensitivity.”

Figure 3

Figure S4

For the purpose of discovering markers of a given cell population, we decided to set $X=0.15|K|$ and $L=2|K|$, where K denotes the cluster of interest and $|K|$ refers to the cardinality of K . This means that when searching for an expression cutoff to binarize gene expression data, we require at least 15% (X) of cells in the cluster of interest to be above the cutoff value, and that the number of cells above the cutoff value does not exceed twice the size of the cluster of interest (L).

We consider the values ($X=0.15|K|$, $L=2|K|$) to be biologically reasonable. We validated on simple examples that they result in desirable properties for the test. In particular, we compared ($X=0.15|K|$, $L=2|K|$) to other values including ($X=0$, $L=N$, where N denotes the total number of cells) corresponding to the basic mHG test (see new Figure 3B and new Figure S4). In these plots (below) it is shown that the basic mHG test does not enjoy these desirable properties, hence X and L need to be picked carefully.

While in this manuscript we focus on parameters $X=0.15|K|$ and $L=2|K|$ because we believe they provide good predictions of surface marker panels, we emphasize that the user can change the values of X and L when running COMET. The specific X and L values chosen do not affect the performance of the COMET algorithm. The choice of the ideal X and L parameters for a given marker-panel-detection problem should be set by the user based on their willingness to tolerate false-positive and false-negatives.

For clarification of the rationale behind the choice of the X and L parameters used by default in COMET we have added the discussion below to the XL-mHG section of the Methods section of the manuscript, page 18:

In COMET X and L are set to $X = 0.15|K|$, $L = 2|K|$ by default. When searching for an expression cutoff to binarize gene expression data, we require at least 15% (X) of cells in K to be above the cutoff value, and that the number of cells above the cutoff value does not exceed twice the size of K (L). We consider the values ($X = 0.15|K|$, $L = 2|K|$) to be biologically reasonable. We validated on simple examples that they result in desirable properties for the test. In particular, we compared ($X = 0.15|K|$, $L = 2|K|$) to other values in simulated data including ($X = 0$, $L = n$) corresponding to the basic mHG test (see Figure 3B and Appendix Figure SS4). In these plots it is shown that the basic mHG test does not enjoy these desirable properties, hence X and L need to be picked carefully. While in this manuscript we focus on parameters $X = 0.15|K|$ and $L = 2|K|$ because we believe they provide good predictions of surface marker panels, we emphasize that the user can change the values of X and L when running COMET. The specific X and L values chosen do not affect the performance of the COMET algorithm, but only the quality of markers discovered by COMET. The choice of the ideal X and L parameters for a given marker panel-

detection problem should be set by the user based on their willingness to tolerate false-positives and false-negatives.

3- "we chose a heuristic core size, N ". What does a core size mean in this concept?

The core size determines how many genes are to be considered in the "reduced sized top-gene set" used in our heuristic to reduce the space over which gene-combinations are assessed. Specifically, this number tells us how many of the top-ranking single-genes will be used to pair with other genes to create our panel search space. For instance, in the 2-gene case a core size of 50 will take the top 50 performing single genes and the heuristic will test all gene pairs in which at least one of the genes in the pair is present in the "core set" of the top 50 performing single genes. In the 3- gene panel case, the core takes that many single genes, creates pairs amongst those genes, then matches those with all genes. We have edited the Methods section of the main text to make this point clearer on page 25. We have added the underlined text:

"In the heuristic approach, we chose a heuristic core size, N , and compute 2-gene or 3-gene combinations. This core size tells us the number of top-performing single genes we will use to pair with all other genes, creating a new search space."

4- It is very surprising that random forest does not work well here. I would suggest to report the error rate of the random forest (out-of-bag error) as well to ensure that your implementation was a good one.

We thank the reviewer for raising this important point. In the submitted manuscript we used the Extra Trees classifier with parameters as specified in the Methods section (a standard parametrization was used). An Extra Trees Classifier is a type of Random Forest classifier which, when splitting at a given node, selects a subset of the features and a random split point for each of these features. The feature whose split point leads to the highest decrease in Gini impurity is chosen by the algorithm.

Following this reviewer's input, we now added the standard Random Forest Classifier to our analysis in the manuscript (Figure 3, Appendix Figures S5 and S6, see below) and to the Methods section (sub-sub-section *Comparison of COMET to classifiers*, sub-section *Simulations, Methods* section, page 28). The same parameters were used as for the Extra Trees classifier (Methods). Appendix Figure S6 is new: in this figure, we utilize a new simulation engine which named the noisy Poisson-Gamma generative model, in order to generate synthetic scRNA-seq data. We implemented this additional simulation engine following reviewer remarks. Further details can be found in the Methods section (page 31). In all cases, we observed that the Random

Forest model performed better than Extra Trees with respect to the SSR score, and that both models performed worse than COMET.

To insure that the Random Forest and Extra Tree models generally achieve good performance, we also included an analysis of the out-of-bag error for both Random Forest (RF) and Extra Trees (XT) as shown below. In both cases we observe low out-of-bag error for both RF and XT indicating that the tree ensemble models used achieve good predicting performance under this measure. We report that the XL-mHG test outperforms XT and RF in our simulations under the SSR metric, which suggests that some poor markers play an important role in RF and XT's good predictive performance. Hence these methods are good for prediction but appear less relevant than the XL-mHG approach for marker detection.

Interestingly, we observe that XT achieves a lower out-of-bag error than RF in our simulations. This is compatible with results from the foundational article on Extra Trees by Geurts et al. (2006): "On classification problems, Extra Trees provides the best results in terms of error rate, although in terms of "bias+variance" of probability estimates it is sometimes inferior to RF".

Figure 3

Figure S5

Figure S6

5- Page 27: SSR sums up the rank of 's' genes and divides that by the optimal sum of ranks to quantify the performance of different classifier techniques. Why don't you measure the absolute difference between the predicted and the optimal ranks?

We recognize that the absolute difference between the predicted and the optimal ranks would be a good option when comparing two gene rankings. However, we opted not to use this metric to assess the performance of classifiers in Figure 3C due to multiple ties within the rankings of each category of our simulated data (good markers, poor markers and non-markers). There is no unique and strict good ranking we can compare to, but rather a “block ranking” in which we would like all of the “good markers” to rank above the rest of the genes, but their specific order does not matter. The SSR score is designed to account for this setting in which many ties are present within the rankings.

We explain why there are multiple ties in our simulated dataset: We generate a cell-by-gene expression matrix including good markers (say for example, G_1 to G_5), poor markers (G_6 to G_{10}) and non-markers (G_{11} to G_{15}). While there is a clear order between these three categories (a good marker is a better marker than a poor marker, which in turn is better than a non-marker), with this generative model all genes within a category are equivalent. There is no difference between ranking G_1 as the 1st, 2nd, or even 10th marker.

Our generative model does not provide a ranking for the genes within each category; furthermore, any data-based ranking would necessarily be method-based. This is the reason why we chose to look at the sum of ranks through the SSR metric: this metric will only be affected if a gene among G_1, \dots, G_{10} has a ranking which is higher than 10. For example:

$$\begin{aligned} \text{SSR}(G_1, G_2, G_3, G_4, G_5) &= \text{SSR}(G_5, G_4, G_3, G_2, G_1) = 1 \\ \text{SSR}(G_1, G_2, G_8, G_3, G_4) &> 1 \end{aligned}$$

For clarity, we made this point more precise in the manuscript, sub-sub-section *Comparison of COMET to classifiers*, sub-section *Simulations*, section *Methods*, page 30: “For example, if \mathfrak{S}_s denotes the symmetric group of $\{1, \dots, s\}$ then for any permutation $\pi \in \mathfrak{S}_s$ the rankings $r_0 = (H_1, \dots, H_{s-1}, H_s)$ and $r_\pi = (H_{\pi(1)}, \dots, H_{\pi(s-1)}, H_{\pi(s)})$ are exactly equivalent. The SSR metric is designed to respect this property (in the example, it gives an equal score to both rankings r_0 and r_π).”

6- "A final rank is assigned to each gene as the average of both XL-mhG p-value rank and the absolute log2-fold-change rank." It is not very clear how this ranking approach works. However, a well-known policy in the optimization domain is that the objectives (here p-

value and log2-fold-change) are normalized first and the weighted sum or subtract of them are used for ranking.

Multiple ranking aggregation procedures can be implemented, as surveyed in Boulesteix and Slawsky (2009). Normalizing the p -value and log2-fold change values, averaging these two quantities and computing a rank is certainly one solution. In this context many solutions well-known to the optimization field require the rankings to be independent. However, the XL-mHG p -value ranking and the log2-fold-change ranking are dependent upon one another.

We opted for the simple solution of obtaining two rankings (one for p -values, the other one for log2-fold change) for the gene list of interest, then aggregating these rankings as the arithmetic average. We found this approach to work well on several data applications. In addition, it is mentioned in Boulesteix and Slawsky (2009), page 9: “a straightforward method to combine several rankings consists of computing a univariate summary statistic for each gene ... e.g. the mean ... The genes are then reranked according to their summary statistic.”

To clarify why we opted for this approach, we emphasized this point in the manuscript, sub-sub-section *Computing and ranking 1-gene marker panels*, sub-section *COMET Algorithm*, section *Methods*, page 22: “We deemed this simple rank aggregation rule (Boulesteix & Slawski, 2009) relevant and parsimonious given that the two rankings cannot be considered independent from one another.”

7- Page 38, "with respect to both robustness to small effect-size". What does "effect size" mean in this concept?

We thank this reviewer for pointing out the need for clarification. We mention in the legend of Figure 3 that “the XL-mHG test outperforms various DE tests in identifying favorable marker genes to be used as markers from simulated data sets, with respect to both robustness to small effect sizes and sensitivity to sample size.” In this context, small effect size refers to a small difference in mean between the cluster of interest and the remaining clusters (in other words, a big overlap between the two distributions).

For clarity we have slightly change Figure 3A (see below) and have extended our description of Figure 3B in the figure legend:

B, left: When varying the magnitude of the difference between the means of the expression distributions for the cluster of interest (K) compared to the background (C) (termed here “effect-size”, see illustration in (A)) common DE tests drop below 0.05 significance level at small effect-sizes (of approximately 0.4), while the XL-mHG test reaches significance only at approximately 3.6. Identification of favorable marker genes requires achieving satisfactory sensitivity and specificity

rates which would not be achievable in cases of small effect sizes (due to the large overlap across the compared distributions). Hence, the XL-mHG test preforms better than commonly used DE tests in that it does not assign significant p-values to genes that are differentially expressed but would be poor markers due to small effect-sizes.

B, right: When varying the total number of cells simulated in clusters K and C (termed here “sample size”, see illustration in (A)) for a fixed and small effect size of 1, common DE tests pick up the small difference in expression as significant once the sample sizes become large (and the detection power increases), while the XL-mHG test does not reach significance and would not consider such genes as potential markers. The small effect-size in this example simulates a poor marker for which desirable sensitivity and specificity rates could not be achieved, and this is controlled in the XL-mHG test by the X and L parameters.

Figure 3

8- Fig. 3 (B): An explanation on how to interpret these plots will help a lot to understand them.

We thank the reviewer for highlighting this point and believe a more extended explanation benefits readers. We have added a broader explanation than before to the legend of Figure 3B, with the two paragraphs noted above in our response to this reviewer’s point (7), and have added the explanation below as a new sub-section *Properties of the XL-mHG test* in the *Methods* section, page 19:

In Figure 3B, we analyze the performance of the XL-mHG test compared to other differential expression tests (Welsh’s t-test, Wilcoxon rank-sum test, Kolmogorov-Smirnov test, Likelihood ratio test for logistic regression). Data is simulated for one gene in many cells across two clusters: the cluster of interest and another cluster including all the remaining cells (Methods).

Figure 3B-left displays, for each test, the corresponding p -value (averaged over several simulation runs) versus the difference in means between the two simulated clusters (the cluster of choice and the background cluster), where the number of cells simulated is fixed. For the classic DE tests, the bigger the mean difference ϵ , the lower the p -value. We observe that the p -values for these tests drop below the 0.05 significance level for ϵ close to 0.4. Note that in this case, the gene is indeed differentially expressed, however it would constitute a poor marker for the cluster of interest as its distribution in the cluster of interest overlaps too much with its distribution in the background cluster. The XL-mHG test, however, only picks up significance for ϵ close to 4. In that case, the expression distributions in both clusters are significantly less overlapping, which makes the gene a good marker candidate.

Figure 3B-right displays, for each test, the corresponding p -value (averaged over several simulation runs) versus the sample size (number of cells simulated from each cluster, where the cluster of interest represents a fixed percentage of the total number of cells, see Methods) for a fixed and small value of the difference in mean between the cluster of interest and the background cluster ($\epsilon = 1$). Note that in this case, the gene is effectively differentially expressed across the clusters. For the classic DE tests, p -values gradually decrease with sample size, since higher sample sizes correspond to higher effect sizes, and hence a higher statistical power for these tests. While the simulated gene distributions are slightly different between the clusters, due to the small size of this difference, $\epsilon = 1$, the gene should not be considered a good marker because it would not achieve good specificity or sensitivity. Fortunately, the XL-mHG p -values do not drop below significance level for this gene regardless of sample size. This suggests that the XL-mHG test will ignore poor markers regardless of sample size, which is a desirable feature, especially when analyzing scRNA-seq datasets which could be very large

9- The authors compared their proposed approach with the related work only using synthetic data. It is interesting to see if COMET still outperforms other approaches for real data as well. For example, in figures 4 and 5, what are the gene-marker panels that previous approaches find?

We agree with this reviewer that it is interesting to see how COMET's predictions on real single-cell RNA-seq data align with predictions made by other methods. We compared COMET's performance in predicting single-gene markers which were validated in Figures 4 and 5 to the other approaches discussed in this manuscript (other methods do not provide predictions for multi-gene markers and hence we restricted our comparisons to the single-gene marker predictions). We

found that when predicting markers for B cells COMET was comparable and slightly better than the other methods, and that when predicting markers for follicular B cells (a cell subtype) COMET outperformed other methods in predicting Cxcr4 and was comparable for the other two markers validated. We give details below.

In the case of identifying single-gene markers for the B cell population in spleen (results shown in Figure 4), the top single-gene markers identified by COMET and validated by flow-cytometry were ranked highly in all methods considered. Interestingly, the B cell marker Cd19 was ranked highest by COMET (rank 2) as compared to the other methods compared to (reaching rank 5 or lower by other methods). The other markers validated in Figure 4, Ly6d, Cd79b and Ms4a1, were all in the top 5-ranking genes by all methods. Achieving a good ranking for B cell markers by all methods is expected given that identifying markers for a distinct cell type is a relatively easy problem, and we were happy to observe that COMET's results in this case align with other methods. We have added an Extended Data Table (Expanded View Table 1) in which we show the COMET output and rankings for the B cell cluster in spleen, along with the rankings generated for each gene by each of the other methods. This is added to the text at page 11: "When comparing COMET's performance to that of other methods for identifying single-gene markers we found that COMET's rankings were slightly higher from that of other methods for the two well-established B cell markers Cd19 and Ms4a1 (CD20), and slightly higher or comparable with respect to the two markers we validated by flow-cytometry (Ly6d and Cd79b) (Extended View Table 1). Having all methods tested be comparable with respect to the identification of single-gene markers for B cell markers by all methods is expected given that identifying markers for a distinct cell type is a relatively simple task."

In the case of identifying single-gene markers for follicular B cells in spleen (results shown in Figure 5), two of the three top single-gene markers identified by COMET and validated with flow-cytometry were ranked highly in all methods considered, while the 3-rd ranking single-gene marker by COMET, Cxcr4, did not rank as highly in all other methods compared to (rankings were 12, 23, 36, 50 and 53 for methods Wilcoxon Rank Sum test, MAST, t-test, Random Forests and Likelihood Ratio Test, respectively). Cxcr4 was validated by us with flow-cytometry as a good marker for follicular B cells, possibly the best marker of the three follicular B cell markers predicted and validated. Additionally, the known marker of follicular B cells, Cd23, ranked 4th in COMET and had a rankings of 3, 4, 4, 10 and 21 for methods MAST, Wilcoxon Rank Sum test, Random Forests, t-test and Likelihood Ratio Test, respectively. Our comparative analysis emphasizes that COMET can predict good single-gene markers for cell subtypes which are not picked up by other methods. We added an Extended Data Table (Expanded View Table 2) in which we show the

COMET output and rankings for the follicular B cell cluster in spleen, along with the rankings generated for each gene by each of the other methods. We have added the following sentence to the text to discuss these results at page 12: “When comparing COMET’s performance to that of other methods for identifying single-gene markers for the follicular B cell cluster, we found that COMET’s ranking of *Cxcr4* was significantly higher than the ranking of any other method (rankings by other methods ranged from 12 to 53, for Wilcoxon Rank-Sum test and LRT, respectively, see Expanded View Table 2), and was comparable for the other validated markers (*Cd55* and *Sell*).”

Importantly, we would like to note that our intention with the analysis and validation presented in Figures 4 and 5 is to show that COMET’s performance on single-gene markers does not fall short from that of other available methods. We thank the reviewer for proposing the analysis provided above, which emphasizes this point and provides an example in which a good single-gene marker for follicular B cells identified by COMET and validated by flow ranks much lower when assessed by all other methods. In Figure 6 we show that COMET accurately predicts two-gene marker panels, an ability not provided by other approaches. Additionally, we would like to emphasize that COMET is packaged as an easy-to-use command line tool and web interface. Taken together, we believe that the favorable results on single-gene markers, the validated ability to identify good two-gene marker panels, and the usability of the COMET tool enable it to be a leading method for the identification of marker-panels from single-cell data.

10- In Fig. 5 (D), why CXCR4 is performing better while being "rank 3"?

The marker prediction by COMET is based on mRNA expression. Due to multiple factors, the correlation of mRNA levels to surface protein levels is not perfect. Examples of such factors are translation efficiency, translation velocity, protein stability and protein transport to the cell surface. These factors are different for different genes explaining why the COMET rank does not perfectly predict the flow cytometry performance. In addition, antibody quality and specificity contribute to discrepancy between mRNA levels and surface protein detection with flow cytometry. Hence, since COMET predicts markers based on mRNA expression, the surface protein expression is expected to not perfectly correlate with the COMET prediction and needs to therefore be validated using flow cytometry.

We thank the reviewer for bringing up this point which we believe is important to clarify. We added the following explanation to the discussion section in pages 14,15: “Due to factors such as mRNA stability, translation efficiency, protein stability and protein transport, we know that this assumption is not always accurate. Additionally, antibody quality and specificity can contribute to

discrepancies between cellular mRNA levels and surface protein detection rates with flow cytometry. As observed in Fig. 5D, the COMET ranks do not always perfectly correlate with performance in flow cytometry (CXCR4 ranked 3rd but performs better than ranks 1 and 2).”

11- Figure S2 (A): How the user should select among the top-ranked panels?

When selecting panels for flow cytometry, the user should incorporate 1. the COMET rank and 2. the availability of established, reliable antibodies. As can be seen in Fig. 6C and D, we observed panels that were not ranked at the absolute top but did have good antibodies available to perform well (CD62L⁺CD44⁻ Rank 22 and CD62L⁺CD55⁺ Rank 38). We therefore believe that antibody availability (or probe availability when looking for panels for e.g. staining) will be a main factor contributing to the selection of panels.

We thank the reviewer for bringing up this point which we believe is important to clarify. We added an explanation about how we selected marker panels to the legend of Figure 4C: “The genes to validate were selected based on availability of trustable antibodies.”

B. Minor issues

1- The quality of Figure s6 is not high enough.

We have improved the quality of this figure.

2- The legend in Figure S8 (C) is not readable.

The Figure S8 legend has been edited to improve clarity

Reviewer #2:

Here, Delaney et al present Comet, a method for combinatorial prediction of marker genes. Having clustered cells from scRNA-seq experiments, one of the most important questions is often to identify genes that can be used to uniquely identify each cluster. Many clustering methods provide such a feature, but the selection is typically carried out by investigating one gene at a time. It is clear, at least from a theoretical point of view, that this may lead to sub-optimal solutions. Thus, COMET could potentially be an important tool for analyzing scRNA-seq data.

The current manuscript is well-written with very few typos and easy to follow. Nevertheless, I have the following major issues:

A. Major issues

1. I had problems installing the software (I emailed about this). Unfortunately, their fix did not work for me as I only have python 3.7 installed and not 3.6. I tried a few different fixes (I am not an expert on python nor on virtual environments) but I was unable to get it to work. Although there is an obvious solution to this issue (which I did not try), I do think that it highlights a potential issue, as time goes by fewer people will have python 3.6 installed and most people would probably prefer if they did not have to install it just to run comet. If it is possible to run with higher versions of python, then I suggest that you add instructions to your installation guide. If not, it would be a good idea to state this requirement up front more explicitly than it is now.

We are very happy that this reviewer was interested in trying the COMET software and are sorry that the attempts were unsuccessful. We feel confident that if we could communicate directly with this reviewer we would be able to resolve the issue and have COMET run on their framework.

The requirement to run COMET with python 3.6 stems from its dependency on the XL-mHG package that was coded independently of COMET by Dr. Florian Wagner and that we use within COMET (Dr. Wagner has been very helpful throughout the COMET development process, see below and in the acknowledgments section of the paper). To specifically address the issue above we have changed the XL-mHG package to work for python version 3.7, and have provided this version of COMET with this review. However, due to an extra C++ compilation procedure of

the python code of the XL-mHG package, our 3.7 compatible COMET code is significantly slower, and we have therefore not incorporated this fix within the general COMET framework. We have reached out to Dr. Florian Wagner, who is the author of the XL-mHG package, and we are planning to work together to enable a fast version of the XL-mHG package which is compatible with python 3.7, and to integrate that version within COMET.

Generally, we would like to note that we are very pleased thus far by COMET's use by the community, and have successfully and rapidly addressed any issues communicated to us privately via email or publicly (<https://github.com/MSingerLab/COMETSC/issues/1>). COMET has been downloaded via the PyPi pipeline over 600 times in the months of June and July (see downloads without mirrors <https://pypistats.org/packages/cometsc>), and we expect that the general usage of COMET via the github link is larger (but is unavailable for tracking). Since the submission of this manuscript, Conor Delaney has received several requests for input to help run the COMET software. Conor has followed up on all requests until success of running COMET was determined, to ensure that COMET could be successfully ran by all users who have reached out. In the process, we have also made several improvements to the COMET code so that common errors are bypassed. Specifically, since the submission of COMET we have:

- Changed the default setting so that the most common error upon installation (that which we thought this reviewer might be encountering) does not need to be addressed.
- Expanded the range of types of input files COMET can take by enabling COMET to take in input files of unparsed 10X format (output of the 10X pipeline count command), both in the command-line version of COMET and the web interface.

We would like to note that we are dedicated to assisting each user in downloading and using COMET, as well as to continue the development of the COMET code and improving its general usability.

2. I then tried the web interface with the Tabula Muris 10X heart dataset (624 cells). After some struggles with the formatting, I got the job to run, but when I downloaded the results, all the files were empty and there was no error message. This job was ID 0bf44897-a554-403a-80d0-92411d8af4ea. It is obviously hard for me to determine what went wrong, but I would recommend that some sort of error message is communicated to the user if this happens.

Again, we are sorry to hear that this reviewer did not get results in the COMET web interface. Unfortunately, since results are erased after 4 days from the COMET server (to ensure space and user data privacy) we cannot trace down the specific run referred to here, and therefore cannot pinpoint what exactly the problem was.

From our experience, when returned results had empty files there was still some problem with the input formatting, but it didn't crash the whole process at the start. To address this issue and have users be more informed, we have changed the COMET interface such that when such errors occur the user will get an "error email", stating that no output was generated and suggesting that we be contacted, rather than an email stating the results are available.

Since the submission of the COMET manuscript, we have made the following additions to the web interface to ease usability:

- If no output is generated (e.g. empty files as happened to this reviewer) the user is notified by email that there is an error.
- 10X files, as they are output in a compressed format from the 10X computational pipeline, can now be input.
- Comma-delimited files are also accepted as input (not just tab delimited)
- We have added additional explanations of the input formats required to the documentation.
- We have made visualizations optional (following point 6 of this reviewer).

As in the command-line version, Conor Delaney has been diligent in following up with users of the web interface if their jobs were not successful.

We have been very happy to witness global usage of the web interface since the submission of this manuscript. We have so far had over 40 jobs submitted and processed by the COMET user interface, from the following locations around the globe:

various locations, USA

Oslo, Norway

Lausanne, Switzerland

Cambridge, UK

Edinburgh, Scotland

Brisbane, Australia

Würzburg, Germany

BGI group, Shenzhen, China

We are dedicated to the continued development and support of the COMET user interface to ensure that the tool is available to a broad range of users.

3. The authors use several synthetic examples to validate their method. Due to the lack of ground truth, this is an acceptable approach, but I do not think that the authors' have used best practice. Instead of using their relatively straightforward simulation strategy, I would recommend that they instead employ Splatter (<https://genomebiology.biomedcentral.com/articles/10.1186/s13059-017-1305-0>) or some other software which can provide more realistic simulations of scRNA-seq data. The methods employed by the authors are unlikely to capture many of the more complex features of real data.

While we looked into using Splatter for our simulations, the software did not enable the precise simulation setting that we were looking for. Specifically, in Splatter one cannot decide which genes are differentially expressed, and so in our simulations we could not know what the ground truth is, a critical aspect of assessing methods performance. Following this reviewer's comment, we reached out to the author of Splatter to enquire if there is any work-around the matter (<https://github.com/Oshlack/splatter/issues/79>). From this correspondence we learned that because Splatter simulations are probabilistic, it is not possible to exactly control which genes are differentially expressed (one can only specify the probability that a gene will be differentially expressed). This was our reasoning for using a simpler (Gaussian) simulation approach to scRNA-seq expression data (as is done in <https://www.nxn.se/valent/2017/6/12/how-to-read-pca-plots>, for example). We recognize that this simulation engine does not capture many important features of scRNA-seq data, however we use it for simple examples to demonstrate the behavior of COMET compared to other methods.

Following the reviewer's input, we opted to design, in addition to the simple Gaussian generative model, a more complex simulation engine that accounts for the technical noise specific to scRNA-seq count data. We provide a description of this model, named the Noisy Poisson-Gamma generative model, in the Methods section (sub-sub-sub-section *Noisy Poisson-Gamma Generative Model*, page 31) as well as in the newly added Appendix Figure S6 (see below). Drawing from recent publications on the noise structure of scRNA-seq data, this parsimonious

model considers both the sampling noise and the efficiency noise that characterize UMI-based single-cell sequencing protocols. Analyses leveraging this new simulation engine lead to similar conclusions (Appendix Fig. S6, see below) as with the simpler Gaussian model above.

Figure S6

The following sections were added to the manuscript text to describe the Noisy Poisson-Gamma generative model and its results.

On page 9:

“Two distinct simulation procedures were used, including a simple Gaussian generative model for gene expression data and a noisy Poisson-Gamma generative model for gene counts data (Methods).”

As part of the Methods section, pages 31-33:

“(ii) *Noisy Poisson-Gamma Generative Model*. We confirm these results using a more complex generative model for single cell-RNAseq data, so as to capture the noise inherent to transcriptomic datasets. UMI-based single-cell sequencing is subject to two main types of technical (i.e., non-biological) variability: efficiency noise and sampling noise (Wagner *et al*, 2019; Grün *et al*, 2014). The efficiency noise captures the global cell-to-cell variation in sequencing efficiency. The sampling noise is related to low detection rates of mRNA molecules in each cell, and is well approximated using Poisson resampling. In the same spirit as in the previous simulation engine, we simulate an $n \times p$ cell-by-gene count matrix (see Appendix Figure S6A, right), where $n = 500$ cells and $p = 1000$ genes. Cells belong either to the cluster of interest K (10% of cells) or the super-cluster C (90% of cells). The gene structure is three-fold: (i) $x\%$ of the genes are not K markers (this set of genes is called \mathcal{N}), (ii) 5% of the genes are good K markers (this set of genes is called \mathcal{G}) and (iii) $z\%$ of the genes are poor K markers (this set of genes is called \mathcal{P} , and for each gene $G \in \mathcal{P}$, G is expressed in a random subset of $q\%$ of cells in K , denoted as $K_{|G}$). Note that $z = (95 - x)$.

The first step of our simulation procedure consists in generating a cell-by-gene matrix of true counts (ground truth) using a hierarchical Poisson-Gamma model (Appendix Figure S6A, left). For each gene g belonging to a particular gene type T (where $T \in \{\mathcal{N}, \mathcal{G}, \mathcal{P}\}$) in a given cell population P (where $P \in \{K, C, K_{|g}\}$), the mean expression μ_g^P of gene g is drawn from a $Gamma(\alpha_{TP}, \beta_{TP})$ distribution. The true transcript count X_{gi} for gene $g \in T$ in cell $i \in P$ is then drawn from a $Poisson(\mu_g^P)$. The matrix $X = (X_{gi})_{g,i}$ corresponds to the ground truth on top of which we will subsequently add technical noise (Appendix Figure S6A, center). For each cell $i \in K \cup C$, an efficiency scaling factor s_i is drawn from a $Uniform([1 - e, 1 + e])$ distribution, where $e \in (0,1)$ is a parameter. For a given gene g , the noisy version of X_{gi} is obtained by resampling $Z_{gi} \sim Poisson(s_i X_{gi})$. The matrix $(Z_{gi})_{g,i}$ represents the observed (noisy) cell-by-gene count matrix that we use in our simulations to compare the XL-mHG procedure to standard classifiers including RF, XT and LR.

Simulations utilize the following parametrization (note that $\alpha_{\mathcal{P}K_{|g}}$ and z will vary across a range of values in Appendix Figure S6). For every gene type T and cell population P , we set $\beta_{TP} = 0.1$. We set $\alpha_{\mathcal{G}K} = 2^5, \alpha_{\mathcal{G}C} = 2^4, \alpha_{\mathcal{N}K} = \alpha_{\mathcal{N}C} = 2^4$. For a given poor marker gene $g \in \mathcal{P}$, we set $\alpha_{\mathcal{P}K_{|g}} = 2^{10}, \alpha_{\mathcal{P}K_{|g}^c} = 2^4$ where $K_{|g}^c$ refers to the complement cell set of $K_{|g}$. We further set $q = 10, e = 0.2$ and $z = 10$. Results are provided in Appendix Figure S6B,C and are similar to the ones obtained using our simpler Gaussian simulation engine. In particular, the XL-mHG procedure outperforms

other classifiers in recovering good markers from the noisy scRNA-seq count matrices (low SSR). This does not prevent RF and XT from being good classifiers as shown by the low out-of-bag error achieved by both tree ensemble methods.”

4. Currently, results are only reported for immune related datasets. However, I would like to see results for other tissues as well (e.g. from Tabula Muris or the mouse cell atlas) to demonstrate that the method generalizes well.

In ongoing work we have observed COMET to work well in identifying subpopulations of malignant cells

* Figure for Referee not shown

5. It would also be useful if the authors could comment on the importance of sequencing depth for the performance of the method in addition to the sample size. In experimental design there is often a trade-off between total number of cells and reads/cell, so it is helpful to know what combination is best for marker gene discovery using comet.

We agree with this reviewer that sequencing depth can affect the accuracy of COMET predictions, as well as any other algorithms (e.g. clustering) ran on the single-cell data. In the work presented in this manuscript and rebuttal we have applied COMET to multiple different data sets and data types, and have seen COMET to perform very well. These datasets were public and were not sequenced at great depth, and hence we believe that COMET is generally robust to a variety of sequencing depths in that it doesn't require extensive sequencing depth to work well. We believe that rigorous characterization of the effect of sequencing depth on the performance of COMET is beyond the scope of this manuscript.

6. Why is it necessary to provide t-sne coordinates as input? I assume that they are used for visualizing the results? If one is only interested in the best genes then this is probably not needed, right? Would it be possible to make this input optional?

We thank the reviewer for this comment which we believe significantly improved the usability of COMET. Indeed, the visualization coordinates (e.g. t-SNE) were only for visualization purposes and are not used in the COMET algorithm. We have therefore changed both the command-line version of COMET as well as the user interface to enable running COMET without visualizations (see below a screenshot of the current user interface, with a box a user can mark for COMET to run without visualization generation).

- 1 File Uploads:** Contains four file upload fields: 'Expression Matrix (Raw 10X Matrix)', 'Cluster File', 'Visualization coordinates', and '(Optional) Gene List'. Each field has a 'Choose File' button. Below these is a checkbox for 'Skip visualizations'.
- 2 Details:** Contains an 'Email' field and a 'Project Name' field.
- 3 GO:** A red button to submit the job.

On the right side, there is a 'Demo' section with a 'Demo' button and text: 'Not ready to submit data yet? Try running our demo!' and 'This demo runs COMET on the splenic CD45+ cells from the Mouse Cell Atlas (Han et al (2018) Cell), available at the MCA website. The demo only processes the first three clusters of the data.' Below this is a 'Formatting' section with a list of instructions: 'Before submitting your job, make sure you use the proper input file specifications: Tab Delimited for all files except genes, which may be submitted as a comma delimited list or a file with one gene per line; Please use text (.txt) files only! If you have CSVs, simply rename the file extension from .csv to .txt; Expression matrix must have genes as rows & cells as columns; Vis / Cluster files should not have headers; Large data sets WILL be downsampled if larger than 3000 cells, soon to be increased.' At the bottom of the formatting section, it says: 'Still unsure about input formatting? Check out this small_input file example! Download our default gene list here.'

7. In the supplement, the algorithm is presented using matrix algebra terminology. For the 3- and 4-gene cases I was wondering if this could be more elegantly represented using 3- or 4-way tensors rather than the current representation?

While we recognize that using tensor algebra to formulate the 3- and 4-gene cases may make sense from a mathematical perspective, it would probably hurt computation speed or memory allocation in our implementation, especially as we are not considering most of the 3- and 4-gene combinations. Specifically, any combination that includes the same gene several times is discarded. For example, in the 3-gene case all combinations of the form AAA and AAB will not be considered. In addition, ABC, ACB, BAC, BCA, CAB and CBA are equivalent and only one of them should be considered. Formulating the problem with matrix algebra as COMET currently does allows us to automatically eliminate many of these redundant cases.

We clarified this perspective in sub-sub-section *Computing and ranking 3-gene marker panels*, sub-section *COMET Algorithm*, section *Methods*, page 24, by adding the following sentence: “We opted for a matrix representation of the 3-gene case for reasons of computational speed and memory”

8. The optimization is carried out with respect to the clear cluster score which is based on true negative calculations. Although this is a perfectly reasonable objective function, one could also think of other choices. In particular, it might be useful to instead consider precision or recall. Would it be possible to use other quantities instead of the TNs for the optimization and do they present any advantages?

We agree with the reviewer that there are many different ways of approaching this optimization of ranks. COMET already calculates the recall as it is identical to the True Positive rate in this case, and given that both true positive and true negatives are being calculated it would be straightforward for a user to calculate the precision (or indeed any value they find informative) by adding a column to the output with this calculation. In addition to this, we formulated the Cluster Clear Score following our ongoing research work in which we are using COMET to identify markers for cellular subtypes. We noticed that many times we are interested in marker combinations that “clean out” entire contaminating clusters, rather than achieving a “global cleaning” across all populations. To bias our ranking to favor such combinations, we defined and integrated the CCS measure. If the user is interested in ranking by some function of the TP and TN measures instead of the CCS, they can do this easily by simple parsing of the output files.

We included a brief elaboration on the usefulness of the CCS in the main text *Methods* section at page 23: “The CCS favors gene combinations that “clean out” specific contaminating

clusters in an attempt to assist in finding solutions for clearing out entire contaminating populations with the marker panel. If other measures of ranking are preferred, users can use the global TP and TN rates reported for each gene panel to construct alternative statistics to rank by.”

Reviewer #3:

This paper contributes another approach to perform differential expression analysis to identify gene sets characteristic for single-cell RNA seq cell type clusters. The authors claim that their work goes beyond mere differential expression analysis and is tailored to panel prediction for functional assays/follow up of discoveries from scRNA.

The main reasons we claim COMET goes beyond mere differential expression analysis are:

- We identify gene panels of multiple genes (Combinations)
- We provide a ranking for the combinations.
- We binarize the data using a new approach applied to single-cell RNA-seq, enabling sensitivity and specificity predictions.
- We provide a unified software package for the above points that is general and fast, and is available as a command-line tool and from a web interface.

We are not aware of other published methods that provide any of the above features.

The novelty and significance of this work is limited for the following reasons.

We respectfully disagree and provide specific explanations below.

A. Major issues

1. The differential expression task is not novel, has been defined since the introduction of single-cell RNA sequencing datasets and a plethora of statistical/computational solutions proposed. For instance, see Luecken and Theis, 2019 for an overview of best practices on this topic.

We agree with this reviewer that there has been a significant amount of work in addressing the differential expression task in single-cell data. Indeed, a wonderful overview of current best practices and next directions is presented in Luecken and Theis (2019) Molecular Systems Biology, as this reviewer notes. The corresponding author of this manuscript, Meromit Singer, is proud to be listed in the acknowledgments section of this referenced review.

In difference from current available methodologies for differential expression, in this submitted manuscript we expand on the current methods and literature for differential expression to provide a new method which is tailored for marker-panel detection, rather than mere differential expression, both in the single-gene and the multiple-gene case. Specifically, we introduce a binarization approach which enables making predictions about the specificity and sensitivity of single-

gene markers. Additionally, we introduce a framework to identify multi-gene marker panels, an attribute that does not exist in current differential expression assessing methods.

2. The authors claim there is a lack of panel prediction approaches. In fact, there are quite a few panel-prediction approaches out there already. The majority of exploratory scRNA sequencing studies that claim identification of novel cell subsets, follow up by defining a panel of FACS compatible markers to validate the novel subsets and describe their function.

These approaches are conceptually the same as the proposed approach here: identify differentially expressed genes that characterize the cell cluster of interest and use the most prominent ones for flow cytometry-based validation. See e.g. Blecher-Gonen et al. 2019, Jaitin et al. 2019, or Villani et al. 2017 (the latter from the Regev lab).

Indeed, papers which have identified novel cell types have accompanied those with identified marker panels. However, the *methodologies* by which such panels were identified were not systematic, did not account for multiple genes in a structured way (in Villani et al multi-gene panels were constructed manually by curation of the single-gene list), and did not provide a general marker-searching procedure.

Below are summaries of the approaches used by the different referenced papers. COMET provides a systematic approach to single- and multi-gene marker panel detection, along with supporting software. Both of these contributions are novel and are not present in any of the publications listed below.

Blecher-Gonen et al. (2019)

- In this study the authors use scRNA-seq analysis of pathogen-challenged lymph node to identify key cell types and pathways.
- They identify all single genes that are significantly expressed by cells from a given cluster compared to all other cells using the PAGODA2 function `getDifferentialGenes` that performs the Kruskal-Wallis test (generalization of the Wilcoxon Rank-Sum test). They consider these genes as potential markers.
- They also use another DE technique in the article, for which they assume that differences in expression are mostly due to changes in the proportion of cells expressing a set of genes, rather than a few cells highly expressing the same set of genes. Gene expression is dichotomized (to 1

if there is more than one UMI, 0 otherwise). For each gene a logistic regression of binarized expression is performed against covariates of interest. Genes with dichotomized expression not associated with library size (using Wald test and Benjamini-Hochberg correction) are discarded.

Jaitin et al. (2019)

- In this study the authors aim at identifying key factors in adipose tissue immune cell remodeling during metabolic disease.
- They perform scRNA-seq on a population of cells and identify 15 broad immune cell groups. They classify these cell groups based on the expression level of differentially expressed genes.
- Differential expression testing is performed for single genes using the Mann-Whitney U test (equivalent to the Wilcoxon Rank-Sum test)

Villani et al. (2017)

- In this study the authors aim at finding markers for subpopulations of dendritic cells (DCs).
- They perform scRNA-seq on DCs, identify clusters of cells, find discriminative single markers per cluster, use the markers to isolate cells and confirm the existence of the clusters through functional analysis.
- To identify single genes that were enriched in each cell cluster, they search for genes whose expression could serve as a binary classifier for cell identity. For each gene, they report the AUC (area under the curve) value, which is equivalent to performing a Wilcoxon Rank-Sum test. To determine the statistical significance of the cluster markers, they used a Likelihood Ratio Test that incorporates the discrete (on/off) and continuous components of single cell gene expression.
- They only consider multiple-gene markers manually, meaning, from manual curation of the single-gene marker lists: "To isolate these cells by flow sorting, we developed a panel incorporating surface markers derived from the set of uniquely expressed genes: FCGR2B/ CD32B for CD1C_A, and CD163 and CD36 for CD1C_B subsets."

3. The study by Delaney et al. demonstrates its differential expression approach by solving a rather simple task, i.e. by recovering a priori known and abundant B cell subsets. Above example studies by Blecher-Gonen et al. 2019, Jaitin et al. 2019, or Villani et al. 2017 make a significantly stronger point by identifying novel and less abundant cell subsets and by functionally validating these in follow up experiments.

In ongoing work we have observed COMET to work well in identifying subpopulations of malignant cells

* Figure for Referee not shown

4. The proposed approach is conceptually one more agnostic differential expression approach that seems not to be tailored for panel prediction since it does not incorporate any specific cue that enriches for markers that are compatible with flow cytometry-based validation.

In order to account for this very idea, COMET is able to take as input a variable gene list that will remove from the data any genes not in this list. In fact, COMET's interface defaults to using a gene list curated to only contain surface markers, thus tailoring the search to only be relevant towards surface-marker panel identification. Furthermore, the approach is generalizable in the sense that a user could easily look at any gene list of interest to a particular problem, whether they be known markers, an unbiased surface marker search, or a list of probes for e.g. in situ visualization the user feels confident in using. Variable gene list input options are incorporated into both the web interface and command line tool. We mention this attribute of COMET in page 10: "Single-cell data for the spleen tissue from the Mouse Cell Atlas (Han et al., 2018) was processed using COMET for a curated list of murine cell surface proteins (Chihara *et al*, 2018) (a default gene list used by COMET unless specified by the user)."

5. The only modest novelty of this work is the incremental methodological contribution of modeling differential expression as an urn experiment, where the binarized gene expression levels follow a hypergeometric distribution under the null hypothesis.

In this work, the XL-minimal HyperGeometric (XL-mHG) test is leveraged to binarize gene expression data. In that respect, we are not exactly modeling differential expression as an urn experiment. The XL-mHG test enables for each gene to find an expression threshold that is used to binarize gene expression data. This threshold comes with statistical guarantees and useful metrics such as the True Positive and True Negative rates. Binarized gene expression data is then used to build a tool that searches for and ranks gene combinations in addition to single genes as potential markers for a given cell cluster. More details on the XL-mHG test can be found in Eran Eden's master thesis at the Technion – Israel Institute of Technology (<http://bioinfo.cs.technion.ac.il/people/zohar/thesis/eran.pdf>).

6. The hyperparameters of the proposed approach seem to be set ad hoc. It would be important to understand the impact of these settings. E.g. what is the sensitivity of the reported results on the threshold definition for gene expression binarization? What is the sensitivity to the X/L settings of the XL-mHG test.

We agree with this reviewer that a more extensive discussion of the values to which the X and L parameters were set to, as well as a discussion of the effect of varying the X and L parameters and the performance of the basic mHG test, constitute important aspects to be discussed in the manuscript. This aspect was also brought up by Reviewer 1 point 2, and we repeat our response to their point below:

When assessing the performance of the basic mHG test, we observe that as expected, the parameters X and L have a significant influence on the performance of the test. The X and L parameters put bounds on the false-negative (X) and false-positive (L-X) counts. The parameters ensure that each gene considered as a marker would provide at least X true-positive calls and at most (L-X) false-positive calls. Hence, the parameters X and L reflect the tradeoff between true and false positive rates in the binarization performed by the mHG test. These parameters prevent COMET from picking up differentially expressed genes that are not relevant as markers due to low specificity or sensitivity.

To clarify the above point, we added the basic mHG test performance to Figure 3b and Appendix Figure S4 (see below). Additionally, we added the following sentence to the XL-mHG test section of the Methods section (page 18):

“Together these parameters provide a flexible trade-off between the sensitivity and robustness of the test, and prevent COMET from picking up differentially expressed genes that are not relevant as markers due to low specificity or sensitivity.”

Figure 3

Figure S4

For the purpose of discovering markers of a given cell population, we decided to set $X=0.15|K|$ and $L=2|K|$, where K denotes the cluster of interest and $|K|$ refers to the cardinality of K . This means that when searching for an expression cutoff to binarize gene expression data, we require at least 15% (X) of cells in the cluster of interest to be above the cutoff value, and that the number of cells above the cutoff value does not exceed twice the size of the cluster of interest (L).

We consider the values ($X=0.15|K|$, $L=2|K|$) to be biologically reasonable. We validated on simple examples that they result in desirable properties for the test. In particular, we compared ($X=0.15|K|$, $L=2|K|$) to other values including ($X=0$, $L=N$, where N denotes the total number of cells) corresponding to the basic mHG test (see new Figure 3B and new Figure S4). In these plots (below) it is shown that the basic mHG test does not enjoy these desirable properties, hence X and L need to be picked carefully.

While in this manuscript we focus on parameters $X=0.15|K|$ and $L=2|K|$ because we believe they provide good predictions of surface marker panels, we emphasize that the user can change the values of X and L when running COMET. The specific X and L values chosen do not affect the performance of the COMET algorithm. The choice of the ideal X and L parameters for a given marker-panel-detection problem should be set by the user based on their willingness to tolerate false-positive and false-negatives.

For clarification of the rationale behind the choice of the X and L parameters used by default in COMET we have added the discussion below to the XL-mHG section of the Methods section of the manuscript, page 18:

In COMET X and L are set to $X = 0.15|K|$, $L = 2|K|$ by default. When searching for an expression cutoff to binarize gene expression data, we require at least 15% (X) of cells in K to be above the cutoff value, and that the number of cells above the cutoff value does not exceed twice the size of K (L). We consider the values ($X = 0.15|K|$, $L = 2|K|$) to be biologically reasonable. We validated on simple examples that they result in desirable properties for the test. In particular, we compared ($X = 0.15|K|$, $L = 2|K|$) to other values in simulated data including ($X = 0$, $L = n$) corresponding to the basic mHG test (see Figure 3B and Appendix Figure SS4). In these plots it is shown that the basic mHG test does not enjoy these desirable properties, hence X and L need to be picked carefully. While in this manuscript we focus on parameters $X = 0.15|K|$ and $L = 2|K|$ because we believe they provide good predictions of surface marker panels, we emphasize that the user can change the values of X and L when running COMET. The specific X and L values chosen do not affect the performance of the COMET algorithm, but only the quality of markers discovered by COMET. The choice of the ideal X and L parameters for a given marker panel-

detection problem should be set by the user based on their willingness to tolerate false-positives and false-negatives.

7. The authors have separate approaches for evaluating the discriminative power of gene sets of varying size (2-4 tuples). This seems cumbersome.

Why do the authors not evaluate regularized classifiers with that efficiently model - possibly a lot larger - gene groups as interaction terms?

We are currently using the CCS score in the scoring of the 2-gene but not 3- and 4- gene combination due to the computational burden of computing this score. We believe it to be useful and are therefore applying it to the 2-gene case. We are actively looking into implementations which would allow the use of the CCS score in a computationally efficient manner in the 3- and 4-gene case.

With respect to adding gene groups as interaction terms to regularized classifiers, a typical scRNA-seq study will include a number of genes ranging from a few hundreds to 20,000. For a study including e.g. $1e3$ genes, we will have on the order of $1e5$ 2-gene interactions, and even more for 3-gene and 4-gene interactions.

It would be impractical for classifiers such as logistic regression and random forests, even if we enforce regularization. One solution could be to impose hierarchical interactions (Bien et al. 2013); however, we are not willing to make this assumption as one of our goals is to also detect 2-gene markers A&B such that A and B alone are not necessarily good markers (as described in Appendix Figure S1).

In addition, we emphasize that classifiers care about label prediction. As a result, if one feature perfectly predicts class label, other features can be discarded even though they may have a decent predicting power (as shown in Figure S1). Hence it will not be possible to obtain a relevant ranking of the different features using Wald test p -values. That is the reason why in COMET we believe that for small (but important) panel sizes, it is necessary to run an exhaustive search (or a heuristic search that performs well compared to the optimal solution, as proposed in the manuscript).

8. The authors use simulated scRNA-seq data for initial benchmarking. The simulation models seem simple. Why did the authors not resort to publicly available scRNA-seq simulation tools (e.g. Splatter by Zappia et al. 2017)?

We agree with this reviewer that our initial simulation approach was simplistic. We looked into using Splatter during the initial preparation of this manuscript but could not do so for the reasons described below. Reviewer 2 raised a similar point in remark 3 and we write the response again below for convenience:

While we looked into using Splatter for our simulations, the software did not enable the precise simulation setting that we were looking for. Specifically, in Splatter one cannot decide which genes are differentially expressed, and so in our simulations we could not know what the ground truth is, a critical aspect of assessing methods performance. Following this reviewer's comment, we reached out to the author of Splatter to enquire if there is any work-around the matter (<https://github.com/Oshlack/splatter/issues/79>). From this correspondence we learned that because Splatter simulations are probabilistic, it is not possible to exactly control which genes are differentially expressed (one can only specify the probability that a gene will be differentially expressed). This was our reasoning for using a simpler (Gaussian) simulation approach to scRNA-seq expression data (as is done in <https://www.nxn.se/valent/2017/6/12/how-to-read-pca-plots>, for example). We recognize that this simulation engine does not capture many important features of scRNA-seq data, however we use it for simple examples to demonstrate the behavior of COMET compared to other methods.

Following the reviewer's input, we opted to design, in addition to the simple Gaussian generative model, a more complex simulation engine that accounts for the technical noise specific to scRNA-seq count data. We provide a description of this model, named the Noisy Poisson-Gamma generative model, in the Methods section (sub-sub-sub-section *Noisy Poisson-Gamma Generative Model*, page 31) as well as in the newly added Appendix Figure S6 (see below). Drawing from recent publications on the noise structure of scRNA-seq data, this parsimonious model considers both the sampling noise and the efficiency noise that characterize UMI-based single-cell sequencing protocols. Analyses leveraging this new simulation engine lead to similar conclusions (Appendix Fig. S6, see below) as with the simpler Gaussian model above.

Figure S6

A

B

C

The following sections were added to the manuscript text to describe the Noisy Poisson-Gamma generative model and its results.

On page 9:

“Two distinct simulation procedures were used, including a simple Gaussian generative model for gene expression data and a noisy Poisson-Gamma generative model for gene counts data (Methods).”

As part of the Methods section, pages 31-33:

“(ii) *Noisy Poisson-Gamma Generative Model*. We confirm these results using a more complex generative model for single cell-RNAseq data, so as to capture the noise inherent to transcriptomic datasets. UMI-based single-cell sequencing is subject to two main types of technical (i.e., non-biological) variability: efficiency noise and sampling noise (Wagner *et al*, 2019; Grün *et al*, 2014). The efficiency noise captures the global cell-to-cell variation in sequencing efficiency. The sampling noise is related to low detection rates of mRNA molecules in each cell, and is well

approximated using Poisson resampling. In the same spirit as in the previous simulation engine, we simulate an $n \times p$ cell-by-gene count matrix (see Appendix Figure S6A, right), where $n = 500$ cells and $p = 1000$ genes. Cells belong either to the cluster of interest K (10% of cells) or the super-cluster C (90% of cells). The gene structure is three-fold: (i) $x\%$ of the genes are not K markers (this set of genes is called \mathcal{N}), (ii) 5% of the genes are good K markers (this set of genes is called \mathcal{G}) and (iii) $z\%$ of the genes are poor K markers (this set of genes is called \mathcal{P} , and for each gene $G \in \mathcal{P}$, G is expressed in a random subset of $q\%$ of cells in K , denoted as $K_{|G}$). Note that $z = (95 - x)$.

The first step of our simulation procedure consists in generating a cell-by-gene matrix of true counts (ground truth) using a hierarchical Poisson-Gamma model (Appendix Figure S6A, left). For each gene g belonging to a particular gene type T (where $T \in \{\mathcal{N}, \mathcal{G}, \mathcal{P}\}$) in a given cell population P (where $P \in \{K, C, K_{|g}\}$), the mean expression μ_g^P of gene g is drawn from a $Gamma(\alpha_{TP}, \beta_{TP})$ distribution. The true transcript count X_{gi} for gene $g \in T$ in cell $i \in P$ is then drawn from a $Poisson(\mu_g^P)$. The matrix $X = (X_{gi})_{g,i}$ corresponds to the ground truth on top of which we will subsequently add technical noise (Appendix Figure S6A, center). For each cell $i \in K \cup C$, an efficiency scaling factor s_i is drawn from a $Uniform([1 - e, 1 + e])$ distribution, where $e \in (0,1)$ is a parameter. For a given gene g , the noisy version of X_{gi} is obtained by resampling $Z_{gi} \sim Poisson(s_i X_{gi})$. The matrix $(Z_{gi})_{g,i}$ represents the observed (noisy) cell-by-gene count matrix that we use in our simulations to compare the XL-mHG procedure to standard classifiers including RF, XT and LR.

Simulations utilize the following parametrization (note that $\alpha_{\mathcal{P}K_{|g}}$ and z will vary across a range of values in Appendix Figure S6). For every gene type T and cell population P , we set $\beta_{TP} = 0.1$. We set $\alpha_{\mathcal{G}K} = 2^5, \alpha_{\mathcal{G}C} = 2^4, \alpha_{\mathcal{N}K} = \alpha_{\mathcal{N}C} = 2^4$. For a given poor marker gene $g \in \mathcal{P}$, we set $\alpha_{\mathcal{P}K_{|g}} = 2^{10}, \alpha_{\mathcal{P}K_{|g}^c} = 2^4$ where $K_{|g}^c$ refers to the complement cell set of $K_{|g}$. We further set $q = 10, e = 0.2$ and $z = 10$. Results are provided in Appendix Figure S6B,C and are similar to the ones obtained using our simpler Gaussian simulation engine. In particular, the XL-mHG procedure outperforms other classifiers in recovering good markers from the noisy scRNA-seq count matrices (low SSR). This does not prevent RF and XT from being good classifiers as shown by the low out-of-bag error achieved by both tree ensemble methods.”

9. It seems that the classifiers are not tuned to avoid overfitting, e.g. by cross validation, i.e. fitting on training data, hyperparameter/model complexity tuning via evaluation on validation data and performance evaluation on test data. This setup allows for identifying suitable RF parameters (and not arbitrarily choosing one parameter set as done currently) and also to identify a regularization strength for the logistic regression (so far logistic regression is used without regularizing the complexity of the fitted model. This is frequently done e.g. by adding an L1 penalty. This lack of proper tuning could explain the result for the classifiers in Fig. 3C, where with sufficiently large effect sizes of the poor markers these are picked by overly complex/overfitted classifiers.

Due to the specific simulation setting that we are using, we are not performing regularization for logistic regression and tree ensemble methods (Random Forest and Extra Trees).

For logistic regression, we consider one gene at a time. We consider the expression of gene G in n cells $X = (X_1, \dots, X_n)$. Cell labels are referred to as $C = (C_1, \dots, C_n)$. We then perform a logistic regression of C against X and an intercept term, then perform a Likelihood Ratio Test (LRT) against the intercept-only model (Methods). This approach has recently been used in Ntranos et al. (2019). Regularization would not be appropriate here since the logistic regression includes only one predictor. The LRT p -values are then used as a proxy for marker potential.

For tree ensemble methods, several parameters can be tuned including number of trees, bootstrap size, number of genes used to split at a given node, as well as stopping criterion. While we initially considered regularization, we decided to set standard values for these parameters (Methods) in order to ensure comparability. Indeed in Fig. 3C, each dot corresponds to feature importance metrics averaged across a number of runs. Setting reasonable default parameter values allow all these runs to be comparable.

For clarity we added the above justification to the main text, sub-sub-section *Comparison of COMET to classifiers*, sub-section *Simulations*, section *Methods* (page 29): “To ensure comparability across simulation runs, random forests parameters were set to the following standard values”.

10. Fig 3B. This result seems not to be favorable for the XL-mHG test. This test requires much larger effect sizes to achieve a significant DE result and does not achieve significance for any of the benchmarked sample sizes.

This figure demonstrates that COMET enjoys desirable properties thanks to the XL-mHG test, compared to other standard Differential Expression (DE) tests. The main goal of COMET is to detect potential markers, which is a different goal from detecting differentially expressed genes.

The genes simulated for the analyses of Figure 3B are poor markers, which one would prefer not to have as predicted markers. Hence, it is desirable that the XL-mHG test not identify those markers as significant.

We clarified the following points in the legend of Figure 3B. In this figure, we analyze the performance of the XL-mHG test compared to other differential expression tests (Welsh's t-test, Wilcoxon rank-sum test, Kolmogorov-Smirnov test, Likelihood ratio test for logistic regression). Data is simulated for one gene in many cells across two clusters: the cluster of interest and another cluster including all the remaining cells (Methods).

Figure 3B-left displays, for each test, the corresponding p -value (averaged over several simulation runs) versus the difference in means between the two simulated clusters (the cluster of choice and the background cluster), where the number of cells simulated is fixed. For the classic DE tests, the bigger the mean difference ϵ , the lower the p -value. We observe that the p -values for these tests drop below the 0.05 significance level for ϵ close to 0.4. Note that in this case, the gene is indeed differentially expressed, however it would constitute a poor marker for the cluster of interest as its distribution in the cluster of interest overlaps too much with its distribution in the background cluster. The XL-mHG test, however, only picks up significance for ϵ close to 4. In that case, the expression distributions in both clusters are significantly less overlapping, which makes the gene a good marker candidate.

Figure 3B-right displays, for each test, the corresponding p -value (averaged over several simulation runs) versus the sample size (number of cells simulated from each cluster, where the cluster of interest represents a fixed percentage of the total number of cells, see Methods) for a fixed and small value of the difference in mean between the cluster of interest and the background cluster ($\epsilon = 1$). Note that in this case, the gene is effectively differentially expressed across the clusters. For the classic DE tests, p -values gradually decrease with sample size, since higher sample sizes correspond to higher effect sizes, and hence a higher statistical power for these tests. While the simulated gene distributions are slightly different between the clusters, due to the small size of this difference, $\epsilon = 1$, the gene should not be considered a good marker because it would not achieve good specificity or sensitivity. Fortunately, the XL-mHG p -values do not drop below significance level for this gene regardless of sample size. This suggests that the XL-mHG test will ignore poor markers regardless of sample size, which is a desirable feature, especially when analyzing scRNA-seq datasets which could be very large.

For clarity, additional indications have been added to the legend of Figure 3B. We also added a detailed discussion of Figure 3B as a new sub-section *Properties of the XL-mHG test* in the *Methods* section, page 19.

11. Fig 3D. comparison ranking from for Random Forest missing (e.g. according to feature relevance)

We thank this reviewer for pointing this out, and have added Random Forests comparisons to Fig 3D (as shown below).

12. Fig 4: on what comparison is the COMET result based? XL-mHG Test for genes contrasting B cells and non-B cells? Are all non-B cell types annotated in the HCA atlas merged into one group of cells for this comparison?

The COMET result is arrived upon by looking at the B cells in contrast to all non-B cells. The non-B cell groups are indeed merged into one group as we explain in the Methods section describing the binarization approach. In deciding a final threshold for expression, the XL-mHG test will find

enrichment based on the number of in-cluster cells above a chosen threshold vs the number of out-of-cluster cells. Since the binarization is decided by this in vs out of cluster decision, it is true that any cell not in the cluster of interest will be merged into one group of cells.

13. Fig4: what is the ranking produced by the other DE and classification approaches compared to in Fig. 3?

We agree with this reviewer that a comparison of the results displayed in figures 4 and 5. We have compared COMET's ranking to that of the other methods compared to in Figure 3 and show that the COMET rankings are better in some interesting cases, and comparable in others. We have added two Extended Version Tables with this information. This point was brought up by Reviewer 1, point 9. For convenience, we provide here the detailed response to this point:

We compared COMET's performance in predicting single-gene markers which were validated in Figures 4 and 5 to the other approaches discussed in this manuscript (other methods do not provide predictions for multi-gene markers and hence we restricted our comparisons to the single-gene marker predictions). We found that when predicting markers for B cells COMET was comparable and slightly better than the other methods, and that when predicting markers for follicular B cells (a cell subtype) COMET outperformed other methods in predicting *Cxcr4* and was comparable for the other two markers validated. We give details below.

In the case of identifying single-gene markers for the B cell population in spleen (results shown in Figure 4), the top single-gene markers identified by COMET and validated by flow-cytometry were ranked highly in all methods considered. Interestingly, the B cell marker *Cd19* was ranked highest by COMET (rank 2) as compared to the other methods compared to (reaching rank 5 or lower by other methods). The other markers validated in Figure 4, *Ly6d*, *Cd79b* and *Ms4a1*, were all in the top 5-ranking genes by all methods. Achieving a good ranking for B cell markers by all methods is expected given that identifying markers for a distinct cell type is a relatively easy problem, and we were happy to observe that COMET's results in this case align with other methods. We have added an Extended Data Table (Expanded View Table 1) in which we show the COMET output and rankings for the B cell cluster in spleen, along with the rankings generated for each gene by each of the other methods. This is added to the text at page 11: "When comparing COMET's performance to that of other methods for identifying single-gene markers we found that COMET's rankings were slightly higher from that of other methods for the two well-established B cell markers *Cd19* and *Ms4a1* (CD20), and slightly higher or comparable with respect to the two markers we validated by flow-cytometry (*Ly6d* and *Cd79b*) (Expanded View Table 1). Having all methods tested be comparable with respect to the identification of single-

gene markers for B cell markers by all methods is expected given that identifying markers for a distinct cell type is a relatively simple task.”

In the case of identifying single-gene markers for follicular B cells in spleen (results shown in Figure 5), two of the three top single-gene markers identified by COMET and validated with flow-cytometry were ranked highly in all methods considered, while the 3-rd ranking single-gene marker by COMET, *Cxcr4*, did not rank as highly in all other methods compared to (rankings were 12, 23, 36, 50 and 53 for methods Wilcoxon Rank Sum test, MAST, t-test, Random Forests and Likelihood Ratio Test, respectively). *Cxcr4* was validated by us with flow-cytometry as a good marker for follicular B cells, possibly the best marker of the three follicular B cell markers predicted and validated. Additionally, the known marker of follicular B cells, *Cd23*, ranked 4th in COMET and had a rankings of 3, 4, 4, 10 and 21 for methods MAST, Wilcoxon Rank Sum test, Random Forests, t-test and Likelihood Ratio Test, respectively. Our comparative analysis emphasizes that COMET can predict good single-gene markers for cell subtypes which are not picked up by other methods. We added an Extended Data Table (Expanded View Table 2) in which we show the COMET output and rankings for the follicular B cell cluster in spleen, along with the rankings generated for each gene by each of the other methods. We have added the following sentence to the text to discuss these results at page 12: “When comparing COMET’s performance to that of other methods for identifying single-gene markers for the follicular B cell cluster, we found that COMET’s ranking of *Cxcr4* was significantly higher than the ranking of any other method (rankings by other methods ranged from 12 to 53, for Wilcoxon Rank-Sum test and LRT, respectively, see Expanded View Table 2), and was comparable for the other validated markers (*Cd55* and *Sell*).” Importantly, we would like to note that our intention with the analysis and validation presented in Figures 4 and 5 is to show that COMET’s performance on single-gene markers does not fall short from that of other available methods. We thank the reviewer for proposing the analysis provided above, which emphasizes this point and provides an example in which a good single-gene marker for follicular B cells identified by COMET and validated by flow ranks much lower when assessed by all other methods. In Figure 6 we show that COMET accurately predicts two-gene marker panels, an ability not provided by other approaches. Additionally, we would like to emphasize that COMET is packaged as an easy-to-use command line tool and web interface. Taken together, we believe that the favorable results on single-gene markers, the validated ability to identify good two-gene marker panels, and the usability of the COMET tool enable it to be a leading method for the identification of marker-panels from single-cell data.

14. Fig 4B: what data is shown? The caption indicates that splenic B cells from the MCA data is shown. Why are not all B cells positive for the canonical B cell marker CD 19?

Figure 4B is showing the entire splenic population while the B cell population is the largest cluster on the graph with the majority of expression of CD19. At the transcriptomic level detected by single-cell RNA-seq, it is many times the case that a good marker would not be detected as expressed in all cells. There are many forms of error and drop-out in single cell analyses that could influence this. Figure 4C shows other markers for this same population, and we can see that for those markers the B cell cluster is not fully covered either.

15. Figs 4B & C: what besides the colors is different on the two subpanels of Fig 4B and Fig4C? Please indicate in the caption.

Figures 4B and 4C show, on the left, a binarized plot of expression for the given gene (red expressed, blue not expressed) based on the COMET threshold found by the XL-mHG test and, on the right, the continuous plot of expression from the raw input. Showing both of these plots gives the user (and paper reader) a better picture of the expression landscape of the given gene. This distinction is now indicated in the caption as such: "COMET plots the expression of a gene across all cells (right) and the binarize values of gene expression following binarization (red expressed, blue not expressed) by the XL-mHG threshold (left)"

B. Minor issues

1. The authors state that COMET assumes a non-parametric model. That is not correct. The gene expression binarization is part of the COMET analysis and assumes a parametric model, rendering the whole approach parametric.

By non-parametric, we mean that our procedure does not make parametric assumptions on the distribution of gene expression, contrary to other methods that model gene expression in order to perform differential expression testing, e.g. zero-inflated Poisson hierarchical models of gene expression.

For example, nonparametric density estimation using Gaussian kernels does not mean that the procedure does not involve any parameters (there is a bandwidth parameter for the Gaussian kernel), but that one does not make parametric assumptions on the distribution whose density is estimated.

This is the reason why we opt to qualify our procedure as non-parametric.

2. The authors qualify markers identified by their approach as "favorable". This qualification seems not adequate in the respective contexts. "cell type specific" or some other similar qualifier seem to be more adequate here.

In this article, a favorable marker is a gene that is potentially a good marker, or equivalently a promising marker. We cannot say with certainty that it will be a good marker, but in light of the COMET results we can say that it will likely be.

For clarity we made this concept more precise in the *Introduction* section of the manuscript, page 3: "(we use the phrase "favorable panel" throughout this manuscript to describe panels that are expected to achieve good accuracy)."

3. Fig 3D what is the difference between XL-mHG and COMET. Both not the same?

The XL-mHG ranking is determined simply by ordering the genes by their XL-mHG test p-value. The COMET rank uses an aggregation technique of both the XL-mHG p-value and the absolute value of the Log (base 2) Fold Change. This approach takes the arithmetic mean of the two ranks to create an averaged rank of the two, which is shown under 'COMET' in Fig 3D. The main text has been edited to clarify this point in the Methods section page 21: "A final aggregated rank is assigned to each gene as the arithmetic average of both the XL-mHG p-value rank and the absolute log₂-fold-change rank. We deemed this simple rank aggregation rule (Boulesteix & Slawski, 2009) relevant and parsimonious given that the two rankings cannot be considered independent from one another. "

4. What implementations were used for the classifiers?

We recognize that this information was missing from the text. All simulations were performed in Python 3.6 using core Pandas, Sklearn, Numpy and Scipy functions. Details were added to the main text, in sub-sub-section *Comparison of COMET to classifiers*, sub-section *Simulations*, section *Methods*, page 29: "sklearn.ensemble.RandomForestClassifier"; "The same parameters were used to train an extra trees classifier (sklearn.ensemble.ExtraTreesClassifier)".

Thank you for sending us your revised manuscript. We have now heard back from the referee who agreed to evaluate your revised study. As you will see below, this reviewer thinks that the study has significantly improved as a result of the performed revisions. S/he raises however a remaining concern, which we would ask you to address in a minor revision. Specifically, the reviewer requests the addition of a short description of the available evidence that COMET works well across a range of datasets.

REFEREE REPORTS

Reviewer #2:

The authors have addressed most of my comments and this has resulted in an improved manuscripts. However, there are a couple of issues that I am still not happy about:

In many cases, I found the authors' answers to my remarks overly verbose and off the mark. For example, when I complain about the web-site not working properly, providing a list of locations that have accessed the website is not an appropriate part of the response. As most scientists these days have a shortage of time, I believe that brevity is a virtue and I hope that the authors can be more succinct in future responses.

The authors argue that the fact that they have used COMET on many different datasets of different sizes and sequencing depths is sufficient evidence that it works well across a range of cases. However, this is not clear from the manuscript and as a very minimum they could report on some of these parameters in their paper and comment on it in the text (either in the main text or as a supplementary table) so that a reader does not have to chase down this information elsewhere.

Response to reviewer #2:

The authors have addressed most of my comments and this has resulted in an improved manuscripts. However, there are a couple of issues that I am still not happy about:

In many cases, I found the authors' answers to my remarks overly verbose and off the mark. For example, when I complain about the web-site not working properly, providing a list of locations that have accessed the website is not an appropriate part of the response. As most scientists these days have a shortage of time, I believe that brevity is a virtue and I hope that the authors can be more succinct in future responses.

The authors argue that the fact that they have used COMET on many different datasets of different sizes and sequencing depths is sufficient evidence that it works well across a range of cases. However, this is not clear from the manuscript and as a very minimum they could report on some of these parameters in their paper and comment on it in the text (either in the main text or as a supplementary table) so that a reader does not have to chase down this information elsewhere.

To address this concern, we have added the following text to the discussion section of the manuscript, page 14:

We observed COMET to work well across a range of technologies and sequencing depths (from an average count of 547 genes per cell in the microwell-seq MCA dataset to an average count of 1,825 genes per cell in the Smart-Seq2 Tabula Muris dataset).

Accepted

20th September 2019

Thank you again for sending us your revised manuscript. We are now satisfied with the modifications made and I am pleased to inform you that your paper has been accepted for publication.

YOU MUST COMPLETE ALL CELLS WITH A PINK BACKGROUND ↓
PLEASE NOTE THAT THIS CHECKLIST WILL BE PUBLISHED ALONGSIDE YOUR PAPER

Corresponding Author Name: Meromit Singer
Journal Submitted to: Molecular Systems Biology
Manuscript Number: MSB-19-9005